


# Euro-Atlantic winter storminess and precipitation extremes under 1.5°C versus 2°C warming scenarios

Monika J. Barcikowska[1], Scott J. Weaver[2], Frauke Feser[3], Simone Russo[4], Frederik Schenk[5], Dáithí A. Stone[6,7], Matthias Zahn[3]

[1]Environmental Defense Fund, New York City, United States

[2]Environmental Defense Fund, Washington, D.C, United States

[3]Institute of Coastal Research, Helmholtz Centre Geesthacht, Geesthacht, Germany

[4]European Commission, Joint Research Centre, Via Enrico Fermi, Ispra, Italy

[5]Bolin Centre for Climate Research, Department of Geological Sciences, Stockholm University, Stockholm, Sweden

[6]Lawrence Berkeley National Laboratory, Berkeley, CA, USA;

[7]Global Climate Adaptation Partnership, U.K.

*Correspondence to*: Monika J. Barcikowska (mbarcikowska@edf.org)

**Abstract.** Severe winter storms in combination with precipitation extremes pose a serious threat to Europe. Located at the south-east exit of the North Atlantic's storm track, European coastlines are directly exposed to impacts by high wind speeds, storm floods and coastal erosion. In this study we analyze potential changes in simulated winter storminess and extreme precipitation which may occur under 1.5°C or 2°C warming scenarios. Here we focus on a first simulation suite of the atmospheric model CAM5 performed within the HAPPI project and evaluate how changes of the horizontal model resolution impact the results regarding atmospheric pressure, storm tracks, wind speed and precipitation extremes.

The comparison of CAM5 simulations with different resolution indicates that an increased horizontal resolution to 0.25° is not only refining regional-scale information, but also improves large-scale atmospheric circulation features over the Euro-Atlantic region. The zonal bias in SLP and wind fields, which is typically found in low-resolution models, is considerably reduced. This allows us to analyze potential changes in regional- to local-scale extreme wind speeds and precipitation in a more realistic way.

Our analysis of the future response for the 2°C warming scenario generally confirms previous model simulations suggesting a poleward shift and intensification of the meridional circulation in the Euro-Atlantic region. Additional analysis suggests that this shift occurs mainly after exceeding the 1.5°C global warming level, when the midltatitude jetstream manifests a strengthening north-eastward. At the same time, this north-east shift of the storm tracks allows an intensification and north-east expansion of the Azores high leading to a tendency of less precipitation across the Bay of Biscay and North Sea.

Regions impacted by the strengthening of the midlatitude jet, such as the northwest coasts of British Isles, Scandinavia and the Norwegian Sea, and over the North Atlantic east from Newfoundland experience an increase in the mean as well as daily and sub daily precipitation and wind extremes and storminess



suggesting an important influence of increasing storm activity in these regions in response to global warming.

## 1. Introduction

International climate policy discussions use annual mean globally averaged temperature targets as the
metric to anchor climate mitigation and adaptation strategies. While useful for climate policy development and implementation, global temperature targets do not explicitly convey the climate impacts that may be felt by society at seasonal and regional scales and hence make it difficult to justify any target as a safe level of warming (Knutti et al. 2015). The recent Paris agreement (*Adoption of the Paris Agreement FCCC/CP/2015/L.9/Rev.1*, UNFCCC 2015) hopes to limit the rise in post-industrial globally averaged
temperature to no more than 2°C, while pursuing efforts toward the more ambitious 1.5°C target. Accordingly, understanding the changes in regional climate as the result of this half a degree difference in these two global temperature levels is important to clarify projected near-term climate change impacts.

In this study, we focus on projected changes in winter storminess and precipitation extremes over the Euro-Atlantic region. The winter climate in the North-Atlantic-European sector is dominated by variations in
mid-latitude westerly winds which determine the position and intensity of storm tracks and thus the pathways of momentum, moisture and temperature transport. Extra-tropical cyclones dominate the redistribution of energy with a net poleward heat transport. They typically form in the region of strong baroclinic activity at the (sub-)polar front of Arctic vs. (sub-)tropical air masses. Stronger pressure gradients are linked to increased storminess and precipitation over central and Northern Europe and less
storms and precipitation over southern Europe and vice versa for weak pressure gradients (e.g. Pinto et al. 2009). Large-scale storminess is dominated by multi-decadal variations in response to a complex interplay of different factors which may lead to changes in storm track position and intensity. The location of the storm track generally changes seasonally in response to solar insolation. Here, changes in the position of the sea-ice front push storm tracks southward while tropical sea-surface temperatures build a barrier in the
south (Shaw et al. 2016).

Owing to its exposure to the direct impact by cyclones from the North Atlantic, weather extremes in this region frequently cause profound socio-economic costs. Heavy rainfall and intense winds are often associated with extratropical cyclones, and may cause flooding and storm surge, damaging infrastructure, industry, agriculture, and forestry. As an example for the North Sea region, extreme wind gusts can exceed
Cat. 3 Hurricane wind forces like during storm Christian/Allan on 28[th]/29[th] of October 2013 with 171 km/h at the German North Sea Coast and 193 km/h over Denmark (von Storch et al. 2014). Hydrological extremes like the coastal as well as inland flooding over the southern United Kingdom during winter 2013/14 (Schaller et al. 2016) are also closely tied to unusual series of low pressure systems including severe storm clusters and persistent rain. Given the large spatial variation in winter European climate
affected by Euro-Atlantic storminess, any effect of global climate change on storminess could profoundly contribute to the associated regional impacts.



Many observational studies on the hydrological cycle in the recent century show wettening tendencies in the northern hemisphere highlighted by annual precipitation increases over large portions of the European continent including Scandinavia and central–eastern Europe. While these tendencies have also been detected in the winter season over most of these regions, they are not present over the southern flanks

(Maraun, 2013) leading to a north-south dipole structure in precipitation anomalies over the European sector. A similar dipole pattern, with positive sign tendencies for the north and negative sign for the south of the continent, were also found in the records of winter extreme rainfall (Donat et al., 2013, Fischer et al., 2014) and river flows (Stahl et al., 2010, 2012). Other studies (Casanueva et al., 2014; Fleig et al., 2015) linked these changes directly to the altered large–scale circulation patterns. Hov et al. (2013) have shown

that the intensification of the winter heavy rainfall in northern and north-eastern Europe is directly associated with the observed poleward shift of the North Atlantic storm track and weakening of Mediterranean storms.

There is however an insufficient understanding of long-term changes in storminess and their drivers (Seneviratne et al., 2012). Records of extreme winds suffer from large inhomogeneities, contributing to

uncertainty of the derived statistics during period of the satellite era (Hartmann et al., 2013; Feser et al., 2014) in addition to spurious long-term trends in global reanalysis data (e.g. Krueger et al., 2013; Schenk and Stendel, 2016). There is however consistency across multiple datasets and medium confidence in a poleward shift of storm tracks since the $2^{nd}$ half of the $20^{th}$ century (Seneviratne et al., 2012). The observed increase in northern hemispheric storminess towards northern latitudes and a decrease southwards during

the past several decades is consistent with the northward shift of storm tracks and their intensity since at least 1970 (e.g. Ulbrich et al., 2009; Lehmann et al., 2011; Hov et al., 2013; Feser et al. 2014). Wang et al. (2009) attributes these changes since 1950 at least partly to external drivers.

Recent efforts to better understand future impacts of global warming on the Euro-Atlantic climate and weather and their extremes such as mid-latitude storminess typically involve an assessment of changes to

various properties of atmospheric dynamics in global climate models (e.g., changes in wind and sea-level pressure variance) under various Representative Concentration Pathway (RCP) greenhouse gas forcing scenarios (Yin, 2005; Lu et al., 2007; Wu et al., 2011; Feser et al., 2015).

Projections of future for annual precipitation indicate an increase for the northern parts and a decrease for the southern parts of Europe. Both global climate model-based (Sillmann et al., 2013; Giorgi et al., 2014) as

well as regional climate model-based studies (Rajczak et al., 2013; Jacob et al., 2014) agree that the strongest increase in the winter heavy rainfall will occur over Scandinavia and eastern Europe. Moreover, Sillmann et al. (2013) have shown that heavy rainfall is projected to increase even in the regions with a mean precipitation decrease (e.g. over the Mediterranean region). Studies analyzing high-resolution, single-model projections (Kitoh and Endo, 2016; Barcikowska et al. 2017) corroborate these results. This bipolar

pattern, with positive tendencies over the northern flanks of central and western Europe and a decrease over southern parts of Europe, has also been found in a multi-model ensemble projection (Donat et al. 2011) for wind speeds.



Projections of future changes in the mid-latitude storms in the Northern Hemisphere indicate remarkable changes, however their features (e.g. spatial patters and intensity) show a strong dependency on the analysis method as well as the generation of the models. Projections based on the ensemble mean of 16 CMIP3 (early 2000s generation) GCMs (Lambert and Fyfe 2006) as well as earlier modeling studies (Lambert, 1995, 2004) suggest a reduced frequency of extratropical cyclones due to a decreased surface meridional temperature gradient over the northern hemisphere. However, this decrease is not spatially uniform as storm activity south of 60° N over the northeast Atlantic and western Europe opposes this tendency, showing an increase in the CMIP3 projections (Leckebusch et al. 2006). Most of CMIP3 and earlier studies (Della-Marta and Pinto, 2009; Pinto et al., 2009b, Bengtsson et al., 2006; Geng and Sugi, 2003; Leckebusch et al., 2006; Pinto et al., 2006) indicate an eastward extension of storminess associated with an increase in frequency of strong storms over the British Isles, the North Sea and north-western Europe. However, Zappa et al. (2013) has shown that the winter storm track's response in CMIP5 (late 2000s generation) projections manifests as a tripolar pattern, with an increase over the British Isles and decreased activity over both, the Norwegian and the Mediterranean Seas.

In most of the modeling applications, the horizontal resolution constrains the ability of GCMs to simulate both the important regional features and the large-scale circulation. So far, the quality of the simulated present climate and thus presumably projections of future climate have improved over time owing to progressing development of GCM's including resolution and representation of the physical process. Nevertheless, present climate simulations in CMIP5 models still suffer from notable biases i.e. on regional scale.

Zappa et al. (2012) has shown that CMIP5-based cyclones are generally too weak and the DJF storm track pattern is too zonal. These deficiencies are associated with the tripolar bias, manifested by negative anomalies over the Norwegian Sea and central-eastern parts of the Mediterranean, and positive anomalies spreading across north-western to central Europe towards the Black Sea. These biases are largely due to the inability of low-resolution models to correctly capture flow-orography interactions and thus correctly represent the tilt of the eddy-driven jet stream over the North Atlantic. Kelley et al. (2011) showed that the increased horizontal resolution in CMIP5 (~200 km) models potentially allowed for a spatial refinement in the simulated geographical pattern and for improvements in the simulated amplitude of precipitation indices. However the resolutions of the CMIP5 GCMs are not sufficiently high to correctly represent daily precipitation extremes (and their changes) and lead to severe underestimations (Sillmann et al., 2013; Van der; Wiel et al., 2016).

Projections downscaled with Regional Climate Models (RCMs) may refine spatial details but will mostly inherit the large-scale circulation biases from the driving GCMs. Therefore, increasing spatial and temporal resolution in GCMs is crucial to improve the representation of the simulated mean climate, weather extremes, and their changes. Studies (Kitoh and Endo, 2016; Barcikowska et al., 2017) employing relatively high-resolution models (~20 to ~50km) have shown much higher skill to capture both large–scale circulation features, spatial features and magnitude of precipitation extremes.   First experimental





simulations at even higher resolution (1-5 km, Kendon et al., 2014; Ban et al., 2015; Lehmann et al., 2015) were capable of projecting changes in heavy rainfall on sub-daily time scales but are usually too expensive to perform.

While it is important to understand the impacts from the worst case emissions scenarios, in order to support

policy relevant mitigation and adaptation strategies as expressed in the Paris agreement, it is also necessary to assess the role of near-term global climate change in anticipating the shifts in regional climate and weather as a function of the 1.5°C and 2°C climate policy goals. However, there is a wide range of global temperature responses and considerable overlap of the CMIP5 models to lower emission scenarios that encompass the 1.5°C and 2°C levels of global warming (Mitchell et al., 2017). As such, teasing out the

relative differences between these two temperature targets is not trivial and requires an alternate modeling strategy that obviates the transient uncertainty with respect to when a given model crosses either the 1.5°C or 2°C threshold (Kalmarkar and Bradley, 2017), mitigates the impact of potential differences in the phasing and amplitude of internal climate variability, and provides enough ensemble members to adequately distinguish the relevant climate change statistics.

The high resolution CAM5 simulations as part of the Half a Degree Additional warming, Prognosis, and Projected Impacts (HAPPI) project provides such a set of model experiments targeted specifically at differentiating the climate response between the 1.5°C and 2°C global temperature levels and their regional implications (Mitchell et al., 2017). The high spatiotemporal resolution of the CAM5 HAPPI experiments are unique in that they allow for a detailed analysis of large-scale changes to North Atlantic storm track

activity and differential impacts as a function of model resolution - a necessary component for studying changes in precipitation and atmospheric circulation on sub-daily time-sales and for the representation of extreme weather events.

The aim of this study is to assess changes in the winter climate and weather extremes over the Euro-Atlantic region, associated with the 1.5°C and 2°C levels of global warming. Our primary focus is here on

the differences between these two temperature levels in the context of the mean and extreme precipitation, winds, and storminess. The availability of high-frequency model output (3 hourly) allows to investigate changes in sub-daily events and also to extract storm tracks using a tracking algorithm (Feser et al., 2014).

The structure of the study is as follows: Section 2 describes the data and explains the methods used in the analysis. The impact of the horizontal resolution on the representation of atmospheric large-scale

circulation is investigated in Section 3.1. The historical runs are validated against observed mean atmospheric circulation and precipitation, as well as high percentiles of daily precipitation in Section 3.2. Section 4 focuses on changes in the mean climate and weather extremes. A summary and discussion follows in Section 5.



## 2. Data and Methods

### 2.1 Data

To assess the importance of the horizontal model resolution, we first analyzed historical runs of CAM5.1 (http://www.cesm.ucar.edu/models/cesm1.0/cam/), provided by C20C+Detection and Attribution Project

(http://portal.nersc.gov/c20c/main.html/). We compare three runs, which cover the period 1979-2005 and are performed at different resolutions. The CAM5-1-2degree run (hereafter CAM5_2, Wolski et al. 2014), CAM5-1-1degree run (hereafter CAM5_1, Stone et al. 2017) and CAM5-1-0.25degree run (hereafter CAM5_0.25, Wehner et al. 2015) are performed at atmospheric horizontal grid distances of 2.5°x1.875°, 1.25°x0.937°, 0.3125°x0.234°, respectively. The 1979-2005 runs use historical values for all forcings

(GHGs, ozone, volcanic aerosol, solar), except land use (set at year-1850), and also except non-volcanic aerosols, which adopt a year-2000 era repeated annual cycle).

Projected climate change impacts on the mean climate state and on extreme weather are investigated based on model simulations with CAM5.1.2 (hereafter CAM5.1.2_0.25) at the highest available ~0.25° horizontal resolution. The simulations are part of the Half a Degree Additional warming, Prognosis, and Projected

Impacts (HAPPI) experiment (Mitchell et al. 2017). The project is designed to provide model output data describing climate and weather changes under 1.5°C and 2°C levels of global warming, as compared to pre-industrial conditions (1861-1880). The design of HAPPI (Mitchell et al. 2017) provides three time slice experiments, using atmosphere-only models, to create large ensembles of 10-year simulations for the present climate (2006-2015) and potential future climate under 1.5°C and 2°C levels of warming (2106-

2115). The two future run ensembles will hereafter be referred to the +1.5°C and +2°C, respectively. Observed forcing conditions include Sea Surface Temperatures (SSTs) and sea-ice (Taylor et al., 2012). SSTs in future scenarios are prescribed by summation of the observed 2006-2015 SSTs and an offset, estimated between decadal-averages of the 2006-2015 period and the projected warmer global conditions for the 2091-2100 period.

The 2006-2015 runs use 2006-2015 values for all forcings (GHGs, non-volcanic aerosols, ozone, volcanic aerosol, solar), except land cover (set at 1850 year). The Pathway 2.6 (RCP2.6) is used to provide the model boundary conditions for the 1.5°C scenario, and a weighted combination of the RCP2.6 and RCP4.5 for the 2°C scenario. CAM5.1.2 currently provides five members for each of the experiments. At the time the research was conducted, the simulations using the CAM5.1.2 model were completed only for future

warming scenarios. Therefore, the historical simulation of CAM5.1-0.25degree for the period 1979-2005 was used to represent the current climate state.

The analysis of the simulated large-scale circulation will be based on monthly means of hydro-meteorological variables for the winter (December, January, February, hereafter DJF) season. The simulated features are compared with reanalysis of monthly pressure at mean sea-level (hereafter SLP),

zonal wind at 850hPa level, and DJF precipitation rates (hereafter PR) for the period 1979–2005. For SLP and wind we use ERA-Interim, provided by the European Centre for Medium-Range Weather Forecasts (https://www.ecmwf.int/en/research/climate-reanalysis/era-interim), at the spatial resolution of ~0.75° x



0.75°. Precipitation is provided by the University of Delaware (V4.01), http://climate.geog.udel.edu/~climate/html_pages/README.ghcn_ts2.html. It is a global gridded land data set, with the 0.5°x 0.5° horizontal resolution. For comparison of the large-scale features, all variables were interpolated on a common 2.5°x2.5° horizontal grid. For our analysis of daily precipitation data, we use E-

OBS (Haylock et al., 2008; http://www.ecad.eu), provided by the European Climate Assessment and Dataset. The dataset contains daily precipitation sums on a 0.25° regular latitude–longitude grid for the period 1950–2015.

### 2.2 Methods

Our analysis of projected climate change focuses on the North Atlantic and European sector (27°-75° N,

80° W–45°E). While most of the analysis focuses on the DJF season, the analysis of storm tracks is extended to the period of October to March (ONDJFM).

Owing to a so far limited data availability, we use the long historical run of CAM5.1 at ~0.25° resolution (CAM5_0.25), which includes the 1979-2005 period, and also a five-member ensemble for the period 1996-2005 (CAM5.1.2_0.25), when referring to present climate. Five-member ensemble simulations for the

1.5°C and 2°C level of warming are referred to as future +1.5°C and +2°C runs, respectively. Statistical significance of differences in the mean DJF climate between future and present climate is tested with the Wilcoxon signed rank test at the 5 % significance-level.

The extreme precipitation analysis is based on the 95[th] percentiles of 3-h and daily total precipitation ratio and return values (RV) for a return period T=10 yrs. The RV were estimated by fitting Generalized

Extreme Values (GEV) distribution by the method of maximum loglikelihood estimation (MLE)  (Coles, 2001; Smith, 2003; Wilks, 2006, Gilleland and Katz, 2014) to a block (seasonal) maxima in the 50-yrs sample of concatenated member runs. Assumptions that our analyzed data is stationary, independent and identically distributed are necessary to fit with a stationary GEV model and are reasonable for HAPPI model outputs (Wehner et al., 2017).

Return values (RV) for a given return period (T) are defined as values expected to be exceeded once per T-years. RVs are estimated as the values corresponding to $[(1 − 1/T)^{'th}$ quantile] of a sample fitted to the GEV model. For example the 90[th] quantile (10 % exceedance probability) is an RV for a T=10-year period. The analysis here focuses on 10-year periods of RVs, because estimations for longer periods (e.g. 50-year periods with an exceedance probability of 2 %) is more prone to sampling errors and biases due to large

uncertainties on the tails given relatively short samples.

The goodness-of-fit to the GEV model is estimated with the Anderson-Darling (A-D) test. The test is a modified version of the Kolmogorov– Smirnov goodness-of-fit-test.  The A-D test gives more weight to the tail and therefore is more suitable for EV distributions analysis. It validates most of the estimations of extreme precipitation for mid-latitude and high latitudes. As expected, approximations for the regions in the

southern parts of Europe and with generally low mean precipitation have shown larger uncertainty.

The analysis of future changes, should involve ideally a comparison of the 2006-2015 and 2106-2115 experiments, which are both based on the historical (2006-2015) SST variability. Unfortunately at the time





of the analysis, the present climate experiments were available only for 1996-2005 period, and thus based on the historical SST variability in 1996-2005.   Therefore, the climate change signal, derived by differentiating the present climate in 1996-2005 and future in 2106-2115, may be obscured by different states of internal SST variations in these decades. To analyze the uncertainty of the derived changes,

associated with different representation of internal variability in compared decadal time-chunks, we applied a bootstrapping approach. We have created a distribution, derived from differences between the future experiment for 2006-2015 and randomly chosen decadal chunks of the 1979-2005 period. The differentiation was reiterated over 1000 times, for both meridional SLP gradient over the North Atlantic and also mid-latitude zonal wind. The gradient SLP was estimated by comparing the maximum in the

vicinity of Azores (10-40° W, 30-50° N) and the minimum in the vicinity of Iceland (10°-40° W, 55°-75° N). The maximum of the zonal wind was chosen for the region 0°-30° W, 50°-65° N.

**2.3 Storm tracks**

Changes in storminess were explored with two measures of daily values during the DJF season. The first one uses high percentiles (95[th]) of daily wind speed. The second is a transient poleward temperature flux at

700 hPa, computed with the daily meridional wind and temperature deviations from the wintertime average. Anomalies were filtered with a 2-10 days bandpass (Butterworth) filter and averaged over the DJF season. Storm tracks were extracted using a tracking algorithm according to Feser and von Storch (2008). The automated tracking approach facilitates the analysis of spatiotemporal variability of cyclones, their lifetime and intensity (Ulbrich et al. 2009; Neu et al. 2013). The algorithm consists of two parts: detection and

tracking. The first part searches for the local minimum SLP and maximum wind speed. Additionally, before tracking, a spatial digital band-pass filter (Feser and von Storch, 2005) was applied to the 3-hourly output of SLP fields to extract mesoscale features of variability.

A storm was identified when a lifetime wind speed maximum exceeded 10 m s−1, and a pressure minimum dropped to 995 hPa and a filtered pressure anomaly of -5 hPa. Only tracks lasting more than 96 hours were

taken into account in order to extract relatively long-lived and intense storms. Cyclones forming at latitudes higher than 60° N were excluded, to align with the purpose of the study which focuses on the European climate.

Seasonal fields of spatial density (SPD) of 3-hourly storm occurrences were accumulated within a 4°x4° grid and weighted by the unit area. Spatial intensity fields were computed by aggregating within 3°x3° grid

boxes the number of 3-h storm occurrences with maximum intensity exceeding certain thresholds. The threshold for the accumulated wind fields is 10 m s−1 and 0.25 mm hr-1for precipitation. Additionally, maximum intensity values were chosen from each 3°x3° grid falling within an area of 9°x9° from the center of the storm. This approach facilitates the analysis of the storm's impact not only in the regions with local maximum but also for the exposed regions within larger distances from the center.



### 3. Simulated winter mean climate and weather extremes

To evaluate the performance of the CAM5 simulation, Figure 1 compares time-average (1979-2005) SLP fields from observations (ERA-I reanalysis) with three CAM5 historical simulations each run at different resolution, where all data sets are interpolated to the lowest data set resolution (2.5°x2.5° lat-lon grid). All

simulations exhibit realistic patterns of the meridional SLP gradient. However, the gradient between the Icelandic Low and Azores High, which characterizes the typical North Atlantic Oscillation (NAO) pattern, intensifies with increasing resolution. The magnitude of the observed SLP gradient is comparable to the simulation at similar horizontal resolution (CAM5_1). The feature is most intense in the CAM5_0.25, which agrees well with stronger westerlies from Greenland towards the British Isles, indicating a stronger

mid-latitude jet stream. Secondly, both CAM5_2 and CAM_1 show a strong positive SLP bias in the subtropical part of Europe and North Africa, and negative bias extending from Iceland towards southeastern Europe and the Caspian Sea, causing the mean ambient flow (Fig 1 contours of differences between CAM5 and reanalysis) over the eastern N-Atlantic and most of Europe to be more zonally oriented when compared to the reanalysis. These deficiencies are also reflected in anomalies of zonal wind (Fig 2)

along the borders of the SLP circulation patterns.

Both, CAM5_2 and CAM5_1, exhibit anomalously strong westerlies extending across Europe from the British Isles towards Turkey. This corresponds with the zonal bias in the ambient flow and pattern of storm tracks, found in the same regions in CMIP5 models (Zappa et al. 2012). This zonal bias in the ambient flow (Fig 1) is strongly reduced in the high-resolution run (CAM5_0.25).

The results presented here indicate that using high-resolution CAM5 simulations in applications to the winter climate over the Euro-Atlantic regions adds considerably better performance than simply spatially more detailed information. At higher resolution, the large-scale atmospheric flow and associated midlatitude jet stream is better represented, both in terms of the pattern and the magnitude. This improvement will presumably leads to a more realistic representation of the midlatitude storm tracks and

associated with them wind and precipitation over Europe.

Figure 3a shows that mean seasonal precipitation in CAM5_0.25 indeed bears a very close resemblance, compared to observations. However the comparison indicates also a much higher magnitude of precipitation over regions with complex orography (up to 1 mm day-1) such as the Alps and the west coasts of Scandinavia and UK. Our comparison of observed (EOBS) and simulated daily precipitation at the same

resolution (~0.25°) also demonstrates very high skill of CAM5_0.25 in simulating precipitation extremes. Figure 3b compares 90[th] percentiles of daily precipitation extremes, indicating that CAM5 skillfully captures the structure and sharp gradients over orographic complex subdomains. Again, in some mountainous regions like the northwest coast of the Balkan Peninsula and south-west coast of Turkey, the simulated values are much higher than the observed ones.

At the same time, it is important to note that constructing a homogenous and high-resolution observational data sets is severely limited over these regions. Thus the differences between these data sets may originate either by the model bias or observational bias (deficient quality or lack of the observations in these





regions). As pointed out for Spain, the differences between different observational datasets may be higher than differences between model simulations and a certain observational dataset (Gómez-Navarro et al. 2012).

Overall, the comparison strongly suggests that high-resolution runs provide more accurate representation of
the winter climate and weather for the Euro-Atlantic sector, where storms play an important role. A correct representation of the storm tracks, governed by the ambient flow, is crucial for capturing the wind and precipitation extremes in the European region, thus in the following section we will focus on the analysis of CAM5 simulations at 0.25° x 0.25° horizontal grid.

**3.1 Impacts of climate warming at the +1.5°C and +2°C temperature levels**

In this section, we investigate climate and weather changes associated with the two global warming temperature levels 1.5°C and 2°C, specified at the Paris climate agreement, and the recently experienced climate. Differences in the forcing between these three sets of HAPPI experiments is confined to different atmospheric forcing and also to the SST offset, which corresponds to the difference between the decadal average of SSTs in the present climate and in projections reaching 1.5°C or 2°C levels of warming. Each of
the experiment includes also internal climate SST variations (e.g. ENSO), happening during the same decadal period, i.e. 2006-2015. Therefore it is expected that the impacts of internal variations will be canceled out while discriminating between the two (e.g. +1.5°C and +2°C) experiments.

In this study the differentiation between the future and present climate is more complex, due to the lack of available output of present climate for the specified period. As an alternative we use the historical dataset
provided for years 1979-2005. The climate in this period is influenced by internal SST variations being at different phase, compared to the 2006-2015 decade. This difference has to be taken into account when discriminating between present and future climate. Therefore, to assess the contribution of different phases of internal climate variations, we will explore the spectrum of differences, which are estimated in reference to different decadal time-chunks of the period 1979-2005. In the following part we will investigate future
changes in the mean winter climate including precipitation and atmospheric circulation over the North Atlantic and Europe.

**3.2 Large-scale atmospheric circulation and precipitation changes**

Here we wish to explore the future response of winter large–scale circulation to the specified levels of global warming. Figure 4a,b depicts differences between the large-scale circulation at the 2°C level of
warming (CAM5.1.2_0.25) and the present climate, derived as the DJF-average over the 1979-2005 period (CAM5_0.25). The pattern clearly resembles the fingerprint of the previously found global warming response, characterized by intensified and poleward shifted meridional circulation cells and corresponding intensification and shift of the westerlies between these cells (Lu et al. 2007; Yin 2005; Bengtsson et al. 2006; Wu et al. 2011). Anomalously intense westerlies (Figure 4b) extend eastward from north of the
British Isles to the north coast of Scandinavia. This feature corresponds well to a strong increase in precipitation in these regions. The maximum change is located at the northwest coasts of the British Isles





(up to ~1 mm day-1) and Norway (~1.2 mm day-1), which are directly exposed to the influence of extratropical cyclones and the associated large quantities of moisture. Precipitation increases slightly over northwestern Europe (France and Germany, up to 0.4 mm day-1). The intensification of the subtropical high (Fig 4b) is accompanied by easterly anomalies at the southern (equatorward) flanks of the anomalous

divergent flow, which reduces precipitation, with a maximum near the center of the anticyclonic anomaly. The anomalies extend eastward and cover most regions of the Iberian Peninsula, with a maximum located along the northwest coast of the region. This region is most sensitive to changes in transported moisture from the North Atlantic. Precipitation in this region may experience up to 1 mm day-1 reduction, which constitutes a difference of more than 38 %, compared to the 1979-2005 period (Fig S2).

At the same time, changes associated with warming at the +1.5°C level are quantitatively and qualitatively different than those at +2°C. Large-scale circulation manifests as very weak intensification of the meridional cells and doesn't indicate a poleward shift of the cells, when compared to the +2°C experiment. Associated with these changes, the midlatitude westerly anomalies are much weaker as seen between the British Isles and Iberian Peninsula, in contrast to the +2°C experiment with intensified westerlies north-

eastward from the British Isles.

The above comparison of winter changes at the two temperature levels suggests an existence of a critical threshold of large-scale winter climate changes occurring after global warming exceeds the +1.5°C warming level. On the other hand it may also reflects the asymmetry in forcing changes for the 1.5C and 2.0°C experiments. The changes associated with warming at 1.5°C level stem from an interplay of a

number of forcings, including aerosol (generally) reductions, while an additional half a degree warming is solely a consequence of $CO_2$ increases and ocean warming.

The difference, estimated between these two levels (Fig 5a), clearly depicts that the additional half a degree warming from the +1.5°C level yields not only a remarkable intensification of the SLP gradient, but also a strong poleward shift of the circulation cells, midlatitude westerlies, and precipitation anomalies. The

estimated SLP differences show statistical significance at 5 % level, with most regions showing non-zero changes (Fig S5). Figure 5 shows that the maximum SLP anomaly is located over the northern part of Bay of Biscay, while reduced precipitation expands north-eastward, through the Biscay Bay, France, southern parts of British Isles, and the North Sea. Drying over the northwest coast of the Iberian Peninsula is even stronger, compared to the difference in reference to the present climate. Therefore, the zone of increased

precipitation is more confined towards the north, covering northern parts of the British Isles, and the Norwegian coast. Here, it is important to note that the estimated future response may be impacted by the differences in spatiotemporal evolution of the underlying North Atlantic SSTs between the 1979-2005 period and the SST patterns derived from the 2006-2015 time period which were used for the future experiments.

The variability of the winter SSTs over the North Atlantic has been shown (Zappa et al., 2012) to have a rather negligible impact on storminess in the CMIP5 models. However, numerous observational and modeling studies (Ruprich and Cassou 2015; Deser and Blackmon 1999; Kushnir, 1994; Battisti et al.



1995; Delworth, 1996; Seager et al., 2000; Selten et al., 1999) have shown that ocean-atmosphere interactions result in strong coupling between spatial variations of SSTs and SLP during winter. The most known signature of that coupling manifests as the NAO or East-Atlantic (EA) patterns, which determine to a large degree atmospheric large-scale circulation and spatial features of storminess in that region.

In other words, the estimation shown above can either be spuriously enhanced or obscured by the impact of different phases of internal climate variability between the two time-slices. To address this issue, we additionally explored the possible outcome of the estimations, using different decadal time slices of the 1979-2005 period. The time-slices were constructed by randomly choosing ten values for the period. The estimations were repeated 1000 times, and shown as a distribution of derived changes in meridional SLP

gradient and midlatitude westerlies (see Methods 2.2 for details).
    The distributions of these differences, computed for both measures between the two different decadal time chunks, are shown in Fig S4 (Supplementary Material). As one would expect, the distributions converge towards the differences (shown in Fig 4) computed between the future period and the whole period of present climate (1979-2005). They show an intensification of the meridional SLP gradient and midlatitude

zonal wind by approximately 3hPa and 0.9m s–1. Nevertheless, the distributions indicate clearly that the impact of internal variability, when using only decade-length chunks of the 1979-2005 period, limits the robustness of the derived changes. The estimations of changes show in more than 10 % of cases a weakening of the SLP gradient and midlatitude westerlies, which defies statistical significance of the result. Therefore, in our analysis (Fig 4), we used a time-average over the whole 1979-2005 period, which should

partly reduce the impact of internal variability. This process, assuming that internal variability in both data sets has equal contribution, should reduce a probability of weakening SLP gradient and zonal wind well below 10 %, corroborating the main outcome of the analysis. Nevertheless, repeating the analysis when the ensemble runs of present climate for the intended period (2006-2015) are available, will methodically cancel out the impact of internal variability existing in both periods (2006-2015 and 2106-2115) and hence

improve qualitatively the estimate of climate change.

### 3.3 Changes in daily and sub-daily precipitation extremes

    In this section we investigate changes in daily and 3-h precipitation extremes associated with an increase in global warming from 1.5°C to 2°C (CAM5.1.2_0.25). Precipitation extremes are measured with 95th percentiles and 10-year return values, derived by fitting a GEV distribution to the HAPPI model outputs.

Figure 6a,b presents the future response derived for 95th percentile of daily and 3-h precipitation, associated with an additional half a degree of warming. The response shows a bipolar pattern, with an increase over the North Atlantic over the northern part of the typical midlatitude storm track region, and a decrease southward, over the region of anticyclonic anomalies. Significantly increased precipitation anomalies extend north-eastward from Nova Scotia through the northwest British Isles towards the Norwegian Sea

and northern Scandinavia. The maximum change is located along the northwest coasts of the British Isles and Scandinavia (up to ~0.2 and 0.24 mm h-1 in 3-h precipitation, respectively), which corresponds well with the derived changes in mean precipitation.





Figure 6 also exhibits a significant (at 5 % significance level) reduction over the Iberian Peninsula, north-western Europe, and southern flanks of the British Isles. The most radical decrease in subdaily precipitation extremes occurs along the northwest coast of the Iberian Peninsula (-0.25 mm h-1) and in the vicinity of Biscay Bay (-0.18 mm h-1). It is worth noting that changes in the extremes of subdaily precipitation are

larger and more significant over larger areas. For example, the local minimum found in subdaily precipitation northwestward from the Iberian Peninsula, is less recognizable in the daily data, which may indicate a smaller impact of storminess to the daily scales, as compared to 3-h data.

The future response in 10-yr return values for subdaily precipitation (Fig 6c), derived with GEV models, is consistent with the pattern derived with 95[th] percentiles and indicates even more radical changes. For

example, the increase over the northwest coasts of the British Isles and north-western Scandinavia reaches up to 0.3 mm h-1. The decrease in the west part of the continent, found in the analysis of the percentiles, covers a larger area and extends more towards the center of the continent.  The magnitude of the precipitation and their changes in the off-coastal areas is usually smaller. Nevertheless, fractional differences (Fig 7) indicate pronounced changes (compared to the 1.5°C level) approaching a 15 %

decrease in the interior of France, over the North Sea, southern Scandinavia, southeast Europe and up to a 25 % increase in the interior of northeastern Scandinavia.

**3.4 Climatology and changes in subdaily wind extremes and storminess**

In this section we investigate future changes in storminess associated with an increase in global warming from 1.5°C to 2°C. Apart from the chosen forcing scenario, additional uncertainty in predictions of future

climate may be also related to the general model performance, known bias in the historical period and the ability to simulate certain features of interest. As such, a validation of the model skill in simulating the long-term climate is not necessarily a guarantee for skillful future projections. However, it is a useful indicator for the model's fidelity to reasonable simulate features of interest. Hence, before analyzing projected changes for the future, we will start the analysis of storminess here by focusing first on the long-

term mean, simulated with CAM5_0.25 for the period 1979-2005.

Here we use three different measures of storminess:  the 95[th] percentile 3-h wind speeds, band-passed filtered transient poleward temperature flux (VT), and density of storm tracks, which are explicitly extracted with a tracking algorithm. All of these measures have certain limitations in characterizing storminess. Measures of wind extremes and transient temperature fluxes will not distinguish the cause of

the changes, e.g. changing frequency or intensity of storms. An application of the Lagrangian approach facilitates extraction of storm tracks and their properties. However potential deficiencies of models in representing realistically storm features (e.g. underestimated intensity) often limit the feasibility of tracking algorithms to construct a representative sample of storms. Thus the robustness of that approach can be limited due to the sampling bias. An interpretation using all three measures facilitates a more complete

description of the present climate and future changes in storminess.

The analysis of the historical run for the period 1979-2005 shows that CAM5_0.25 reproduces the spatial features of storminess very realistically compared to the observational-based data sets. For example, a



strong meridional tilt is skillfully captured in all three measures (Fig 8b, Fig 9a and Fig S6). For VT (Fig 8b), not only the spatial pattern but also the intensity agrees remarkably well upon direct comparison with the observational climatology (http://www.met.reading.ac.uk/~swrshaff/sstanom.html). The VT pattern manifests the full spatial spectrum of the location of extratropical cyclone activity. The pattern spreads

across the subtropical and midlatitude North Atlantic, featuring maximum values along the region from Newfoundland, across the east Atlantic between the British Isles and Iceland, to the Norwegian Sea. The simulated maximum intensity of VT yields approximately the value of 25 °C m s$^{-1}$, which is very close to the derived values from the ECMWF Reanalysis. The simulated intensity with CAM_0.25 is much more realistic in comparison with one of the CMIP3 models

(http://www.met.reading.ac.uk/~swrshaff/sstanom.html), with typically much lower horizontal resolution. In the latter, the strength of the storm intensity was found to be nearly half of the observed one.

For high wind speed percentiles (Fig S6), which have been widely used (e.g. Krueger et al. 2013) as a simple measure of storm activity, CAM5_0.25 reproduces the pattern of local maximum very closely to the one found in VT. The simulated intensities also bear a close resemblance to the wind extremes (not shown)

in reanalysis data *i.e.* ERA-Interim and CFSR. CFSR, which has the finest (~ 0.25°-0.5°) horizontal resolution, shows a much better agreement with the model. The ERA data set shows lower values than CFSR, especially over the vicinity of the local maximum. The apparent difference stems most likely from the underestimation of midlatitude extreme winds in ERA-Interim and ERA-40, which appears to be related to their relatively coarser spatial and temporal resolution (Chawla et al., 2013; Pielke, 2002; Stopa and

Cheung, 2014; Sterl and Caires, 2005; Campos and Guedes Soares, 2017). As in the case of precipitation mentioned previously, this points again towards the finding, that differences between different observational data products may be as large or even larger than deviations of climate simulation relative to a certain reference dataset (Gómez-Navarro et al. 2012). A coarse model resolution is however not the only explanation for too low wind speeds for high wind percentiles. As e.g. shown by Rockel and Woth (2007),

even regional climate models simulate too low wind speeds for high percentiles if no gustiness correction is applied to the model output.

The climatology of the spatial track density (Fig 9), derived from the tracking algorithm, agrees reasonably well with the tracks gleaned from observations (Zappa et al. 2012, Hodges et al., 2003). However, the pattern in CAM5_0.25 exhibits a maximum shifted towards the Norwegian Sea and does not manifest a

strong activity in the region south-east of Greenland. This feature is well captured in the CAM5_0.25 wind speed percentiles and is most likely associated with the short-lived katabatic winds that descend from the Greenland ice sheet. These features are however not of interest for our analysis and the tracking algorithm used in this study is tailored to extract only the long-lived and most intense cyclones. Upon visual inspection it can also be suggested that the track density simulated in CAM5_0.25 is improved, as

compared to the low-resolution CMIP3 and CMIP5 models. The CMIP models have been shown to exhibit a very strong zonal bias with positive anomalies in central Europe and negative values over the Norwegian Sea (Zappa et al. 2012). We note however that the verification whether increasing the resolution improves





the simulated climatology of midlatitude storms demands further analysis. This would require a unified methodology, with the same tracking algorithm applied to all of the datasets. Overall, first results shown here indicate that the CAM5_0.25 reproduces features of storminess considerably more realistically than coarse resolution simulations, both, in terms of spatial pattern and intensity. This increases our confidence

in the skill in projections of future storminess projected with the CAM5_025, and is the focus of the remainder of this section.

Figure 8a depicts differences in the response between 1.5°C and 2°C level of warming, derived for the 95[th] percentile of 3-h wind speed. The derived changes show a bipolar pattern, similar to the one found for extreme precipitation. The most radical decrease in wind speeds manifests at the poleward fringe of the

subtropics (40° N), between the Iberian Peninsula and the Azores. This region overlaps well with the location of maximum easterly anomalies at the southern flanks of the winter anticyclonic anomaly, found in the analysis of changes in general atmospheric circulation (see Fig 5). Thus it is likely that the simulated reductions in extreme winds are to a large extent caused by the poleward shift of the large-scale circulation, the signature of which is the weakening of the westerlies at the poleward flanks of the subtropics.

The response to the additional half a degree warming is also expressed as a remarkable increase in extreme winds over the northern half of the typical storm track region, with the maximum located between Iceland and the British Isles and along the Scandinavian coast. This feature is highly consistent with the response pattern derived for VT (Fig 8b). Changes in VT indicate a pronounced intensification of storminess on the poleward flanks of their DJF climatology, again featuring a maximum between Iceland and the British Isles

and an eastward extension along the Scandinavian coast. Small negative anomalies occur over the Norwegian Sea, north-east of Iceland. A similar response is found in the storm track density (Fig 9a), showing an increase over the eastern North Atlantic and negative anomalies north-east of Iceland. Positive anomalies found in all measures of storminess collocate well with the local maximum of the intensification of the mean DJF westerlies (Fig 5), which is consistent with the eddy-driven nature of the midlatitude jet

stream.

The analysis of the intensity accumulated along the extracted tracks provides further insights. Figure 9b shows an increase in the number of days, which exceed certain thresholds of precipitation and wind (0.25 mm h-1 and 10 m s-1, respectively). The derived pattern shows similar features to those in the track density except that the positive changes are extended north-east of the Norwegian Sea. An additional analysis (not

shown), repeated for higher thresholds of wind and precipitation, confirms the previous results in that it also exhibits an increase along the Scandinavian coast, indicating that the pattern becomes more zonal for higher intensities.

Overall, the increase manifested in the track density fields over the eastern North Atlantic, between the British Isles and Iceland, is consistent with the anomalies in VT. This suggests that the change in storm

activity in this region is influenced by the increased frequency of storms. The increase in VT and in the number of high intensity days (as diagnosed from wind and precipitation) becomes clearly visible also over the Norwegian Sea, despite no tendencies in track density in this area. For the increased thresholds of the



intensity, positive anomalies emerge also at the coastal regions of Scandinavia, which are also accompanied with insignificant or zero tendencies in track density. Therefore the response found in storm activity over the Norwegian Sea could be alternatively explained by an increase of the intensity of the storms, rather than frequency. This is however a subject for a separate and more thorough analysis. It is also important to note

that the storm tracks analyzed here exhibit a very strong year-to-year variability. Thus the statistics derived here may suffer from large uncertainty, and should be repeated when a larger number of ensemble simulations becomes available, in order to facilitate a reduction in the sampling error.

## 4. Summary and Discussion

In this study we assess near-term regional winter climate and weather changes over the North Atlantic

Ocean and Europe associated with the 1.5°C and 2°C levels of global warming. The design of most state-of-the-art experiments, e.g., Coupled Model Inter-comparison Project (CMIP), are not well suited to address questions on climatic changes associated with the specific climate policy goals. This is due to the fact that CMIP experiments are set in the framework of responses to the particular concentration scenarios, rather than to the particular level of warming. Therefore, we use here a set of ensemble simulations

provided by the HAPPI project. The design of that experiment reduces the impacts of different phases of climate variations and thus facilitates differentiation of the climate response between the two warming levels. The CAM5 simulations provide a set of future climate experiments, describing the global climate and weather at ~0.25° horizontal resolution and at subdaily time-scale (3-h). Hence these simulations create a unique opportunity to explore changes and physical linkages between them across spatial and temporal

scales. Additionally, a set of CAM5 historical simulations, provided at different horizontal resolutions, facilitates an insightful analysis of the benefits of increasing horizontal resolution in regional climate applications.

In the first part of our manuscript, we focused on the assessment of the model's ability to realistically represent key features of winter climate and weather over the Euro-Atlantic sector. Our analysis of the runs,

performed at horizontal resolutions ranging from ~2° to 0.25°, has shown a substantial improvement in simulated large-scale circulation, specifically the meridional SLP gradient and midlatitude zonal winds. The zonal bias of the ambient flow over the North Atlantic and Europe, common for low resolution CMIP3 and CMIP5 (Zappa et al. 2012) models, has been very clearly reduced with the highest model resolution. To a large extent, the reduction of the zonal bias may result from a much better skill to capture ambient

flow-orographic interactions in the model with finer horizontal resolution, suggesting an important upscaling-impact of regional scales in shaping the large-scale circulation.

In the second part of the manuscript, we investigated near-future changes, associated with global warming at the temperature levels specified by Paris agreement. The pattern of the future response, when 2°C warming is compared to the present climate, confirms typical fingerprints of climate response. These are

characterized by a poleward shift and intensification of the meridional circulation cells, manifested here as strengthening meridional SLP gradient, and poleward strengthening and eastward extension of midlatitudes





(Lu et al., 2007; Yin, 2005; Bengtsson et al., 2006; Wu et al. 2011; Feser et al. 2015).

However, different to previous studies, our analysis didn't identify a local maximum of anticyclonic SLP anomalies over the central Mediterranean. This feature was found in many CMIP3 and CMIP5 simulations (Giorgi and Lionello, 2008; Giorgi and Coppola, 2007; AR5, IPCC 2007) and was often used as an
explanation (Giorgi and Lionello, 2008) for reduced precipitation in most parts of this region. Instead, in our analysis, the center of the anticyclonic anomaly is shifted north-westward, which locates it over the North Atlantic, north-westward of the Iberian Peninsula. This feature corresponds well with the shift in drying anomalies, which extend from the eastern North Atlantic and covers only western parts of Mediterranean.
The reason for this difference may be associated again with a strong positive bias in SLP over the Mediterranean and associated zonal bias of ambient flow, persisting in most of CMIP3 and CMIP5 models. Thus, the maximum of the SLP field over the Mediterranean might be partly an expression of that bias. Increasing horizontal resolution to ~0.25° reduces the SLP bias almost completely, as shown in our analysis, which might explain the difference in the response pattern.  In contrast to this result, other
simulations using a ~0.5° horizontal model resolution (Barcikowska et al. 2017) indicated a strong anticyclonic intensification and drying over most of the Mediterranean, despite remarkable reduction of the bias. Therefore, the explanation of this difference in the projected pattern may have other/or additional causes and demands further exploration running different models at different resolutions.

Our analysis also provides additional insights into the evolution of the response, as a function of changing
global temperature and suggests that the poleward shift and intensification of the meridional circulation cells and midlatitude westerlies occurs mostly during the additional half a degree of warming beyond the 1.5°C level. The difference in the response between 2°C and 1.5°C levels is shifted poleward, compared to the changes estimated between 2°C and present climate. The maximum anticyclonic SLP anomaly is located over the Bay of Biscay, which corresponds well with strong relative drying in this region. These
drying anomalies extend also further north-east towards the North Sea, shifting the borderline between opposite sign tendencies northwards. Maximum precipitation anomalies occur in the northwest parts of British Isles, along north-west coast of Scandinavia and the Norwegian Sea.

The evolution of the future response shows a much stronger and distinct pattern compared to the changes prior to the 1.5°C level of warming. This amplification in the change may hence be a reflection of possible
nonlinearities in the response. These could be caused by Arctic amplification and sea-ice loss (Collins et al., 2013), a faster loss of land snow and/or a partial collapse of ocean convection (Drijfhout et al., 2015; Yin et al., 2009) and related changes in ocean circulation.

The response found here of winter weather over the North Atlantic and Europe is largely consistent with the changes found for the mean climate state and large-scale circulation. An increase in warming from
+1.5°C to 2°C level suggests a poleward intensification of daily and subdaily extreme wind and precipitation. These tendencies show the most pronounced impact in the regions most exposed to the inflow of moisture from the North Atlantic, e.g., the British Isles and northwest Scandinavia, where the 95[th]





percentiles of 3-h precipitation increase up to 0.2 mm h-1 and 0.24 mm h-1, respectively. The response pattern derived from daily precipitation shows a very similar pattern to the one derived from 3-h data. However, the latter exhibits larger magnitudes and encompasses larger areas with significant changes. Changes derived with GEV approximations, indicating even more radical shifts, show an increase in 10-yr

return levels of up to 0.3 mm h-1 in the coastal regions of British Isles and North-west Scandinavia. The magnitude of changes in precipitation is smaller in the inland areas. However, many regions like northeast Scandinavia, may still be strongly impacted by an increase of up to 20 %, when compared to the 1.5°C level. Consistent with changes in the mean precipitation along the southern coast of Scandinavia, the east coast of the British Isles and North Sea indicate a slight decrease. These tendencies are more intense and

expand towards western Europe, exhibiting an up to 15 % decrease over France and exceeding a 25 % decrease over the interior and eastern Iberian Peninsula.

Derived changes in extreme precipitation and wind correspond well with changes in storminess, measured here with the transient poleward temperature flux (hereafter VT) and features of explicitly extracted storm tracks. The projected future response, derived from subdaily VT and from spatial density of the

extratropical storm tracks, indicates an increase of storm activity towards the northern side of the current storm track (between Iceland and the British Isles) but also a decrease north-east of Iceland.

The decrease in storminess at the northern flanks of the storm track, measured as the frequency of intense storms, has been identified in the CMIP5 projections (Zappa 2013). Similar to our analysis, the future response according to CMIP5 models suggests a polar amplification of global warming, associated strongly

with the Arctic sea-ice loss. This in turn reduces the lower atmosphere meridional temperature gradient and also baroclinicity, shown here by the decrease of zonal wind northeastward from Iceland, which is consistent with the reduced storminess in this region. At the same time, the minimum of warming SSTs over the North Atlantic could lead to increased surface atmospheric baroclinicity (Brayshaw et al., 2009; Woollings et al., 2012) and thus enhance storminess over the eastern North Atlantic.

An increase of transient poleward temperature flux is found also over the Norwegian Sea, along the Scandinavian coast, which collocates well with the local maxima of increase in extreme precipitation and wind.  The density of storm tracks doesn't indicate any spatially coherent tendencies in this region. However, the positive tendencies in this region emerge when the extreme precipitation and wind events, associated with the extracted storm tracks are analyzed. In these regions we found an increase in frequency

of 3hrly storm occurrences with exceptionally high intensities. The strength of this tendency increases with the intensity of the extreme event, which suggests the possibility of increased frequency of more intense storms. These results should however be confirmed by a more elaborate analysis, specifically targeting changes in storms, and is the subject of further research.

**Acknowledgements:** The authors are grateful to Ángel Muñoz and Alex Petrescu for helpful discussions; and to Dáithí Stone and Michael Wehner for providing data. M.Z. was supported through the Cluster of Excellence "CliSAP" (EXC177), Universität Hamburg, funded through the German Science Foundation



(DFG). DAS was supported by the U.S. Department of Energy, Office of Science, Office of Biological and
Environmental Research, under contract number DE-AC02-05CH11231.

**"The authors declare that they have no conflict of interest."**

**Submission for a Special Issue in Earth System Dynamics: The Earth system at a global warming of
1.5°C and 2.0°C**

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

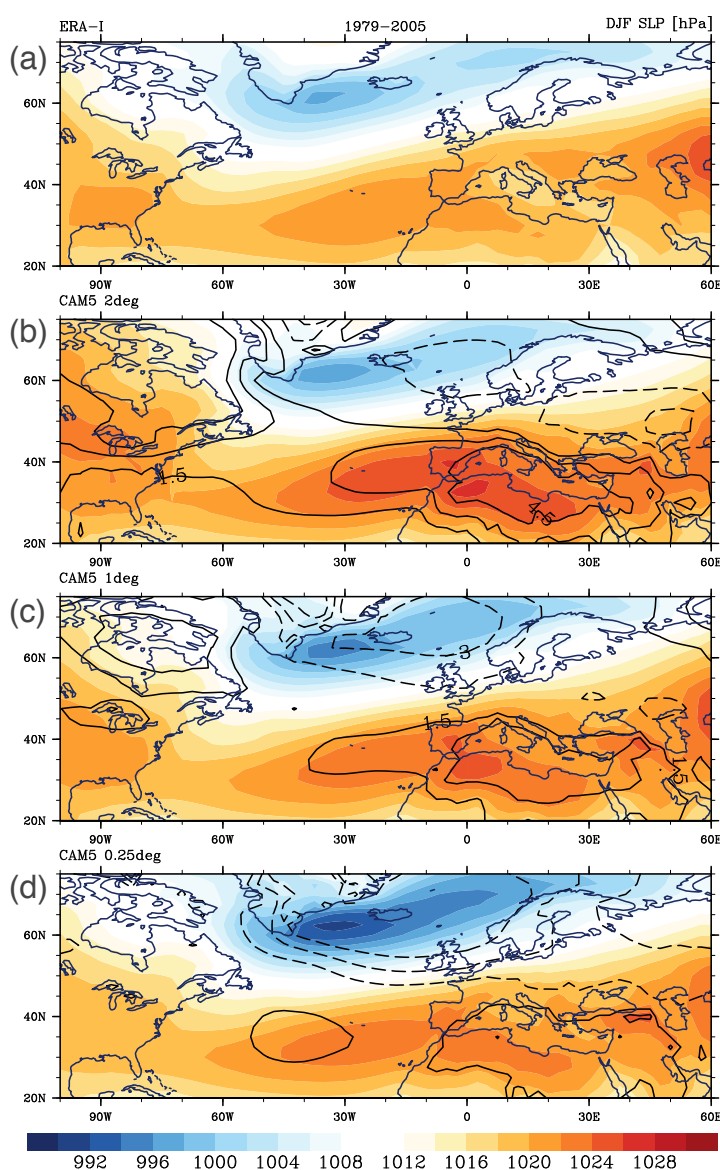

**Figure 1. Time-mean average of the DJF sea level pressure [hPa] over the period 1979-2005, regridded to 2.5° x 2.5° horizontal grid for ERA-Interim (ERA-I,~0.75° lat-lon original resolution), and CAM5 at ~2° (CAM5_2deg), ~1° (CAM5_1deg), ~0.25° (CAM5_0.25deg) lat-lon resolution. Contours show a difference, in reference to ERA-Interim.**



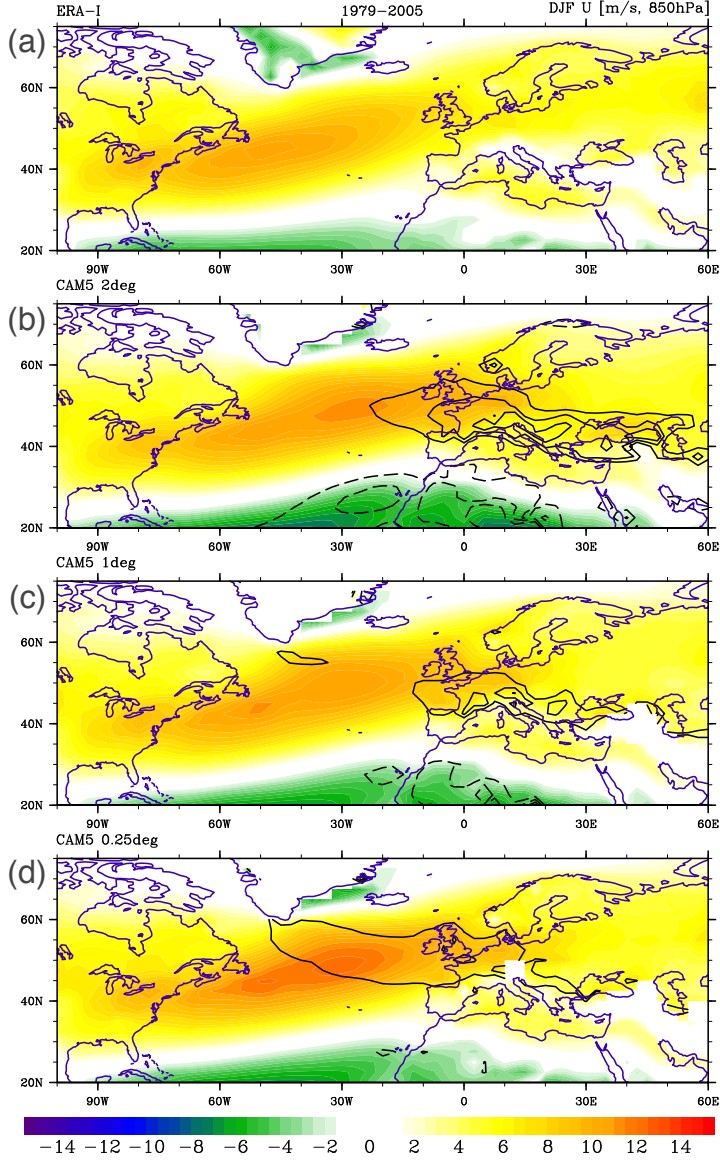

**Figure 2. Time-mean average of the DJF zonal wind [m s–1] over the period 1979-2005, regridded to 2.5° x 2.5° horizontal grid for ERA-Interim (ERA-I,~0.75° lat-lon original resolution), and CAM5 at ~2° (CAM5_2deg), ~1° (CAM5_1deg), ~0.25° (CAM5_0.25deg) lat-lon resolution. Contours show the difference, in reference to ERA-Interim.**





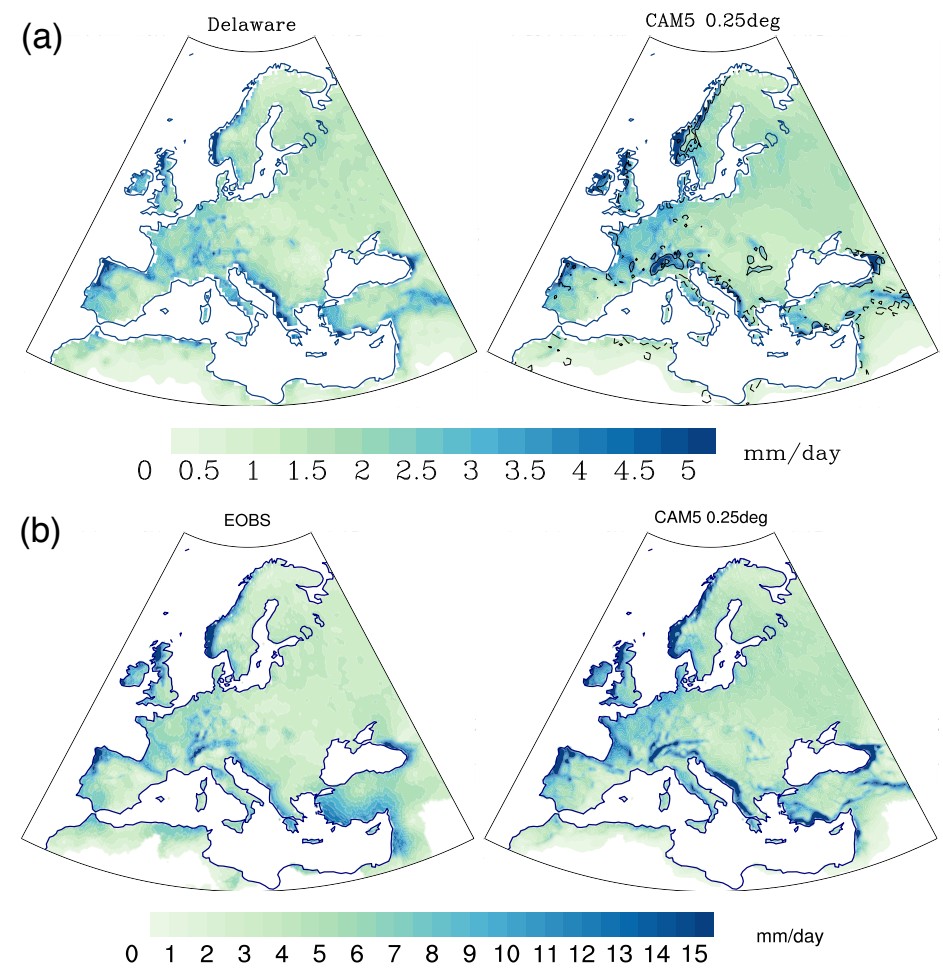

**Figure 3. a) Time-mean average of DJF monthly means of precipitation ratio [mm day-1], averaged over period 1979-2005, in observations (Delware, 0.5° resolution); and the CAM5_0.25 model, smoothed to 0.5°x0.5° horizontal resolution; b) DJF daily precipitation 90th percentiles [mm day-1], averaged over period 1980-2005 in observations (EOBS, 0.25° resolution) and CAM5_0.25 model.**



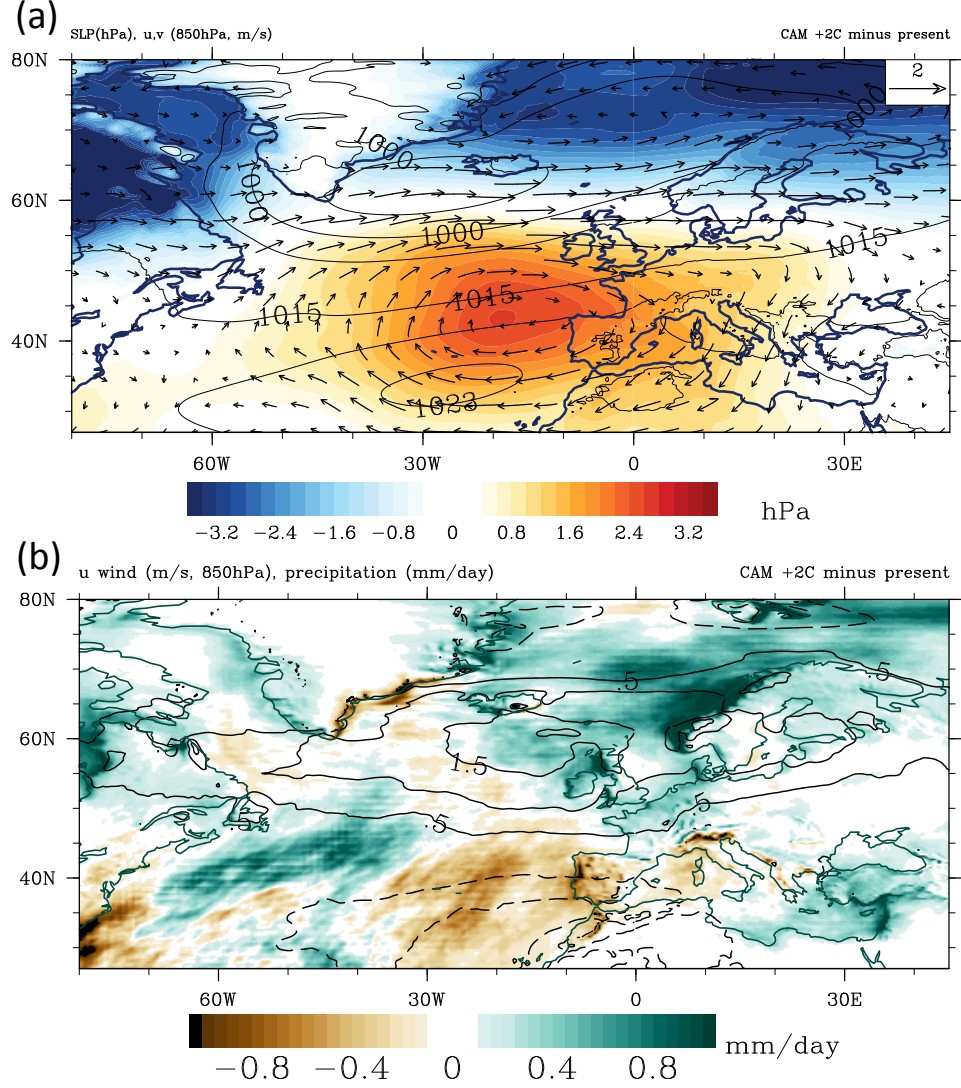

**Figure 4.** Difference between +2°C ensemble and present climate 1979-2005 historical run in DJF a)
sea level pressure [shaded, hPa] and wind vector at 850hPa [m s–1]. Contours show DJF sea level
pressure in present climate 1979-2005, with a local maximum in the vicinity of Azores and minimum
in the vicinity of Iceland; b) precipitation [mm day-1] and zonal wind [contours, m s–1] in
CAM5_0.25.




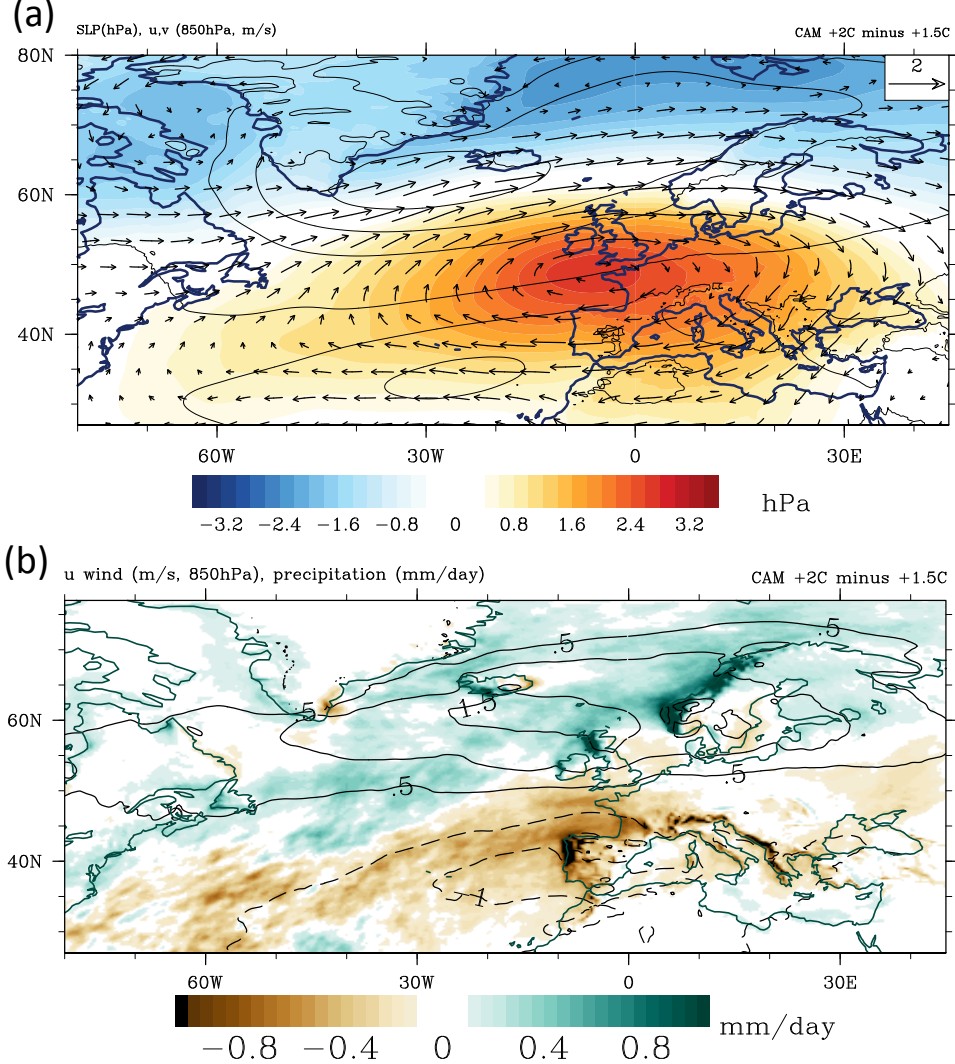

**Figure 5. Difference between +2°C and 1.5°C ensembles in DJF a) sea level pressure [shaded, hPa] and wind vector at 850hPa [m s–1]. Contours show (as in Fig 4a) DJF sea level pressure in present climate 1979-2005, with a local maximum in the vicinity of Azores and minimum in the vicinity of Iceland; b) precipitation [mm day-1] and zonal wind [contours, m s–1] in CAM5_0.25.**





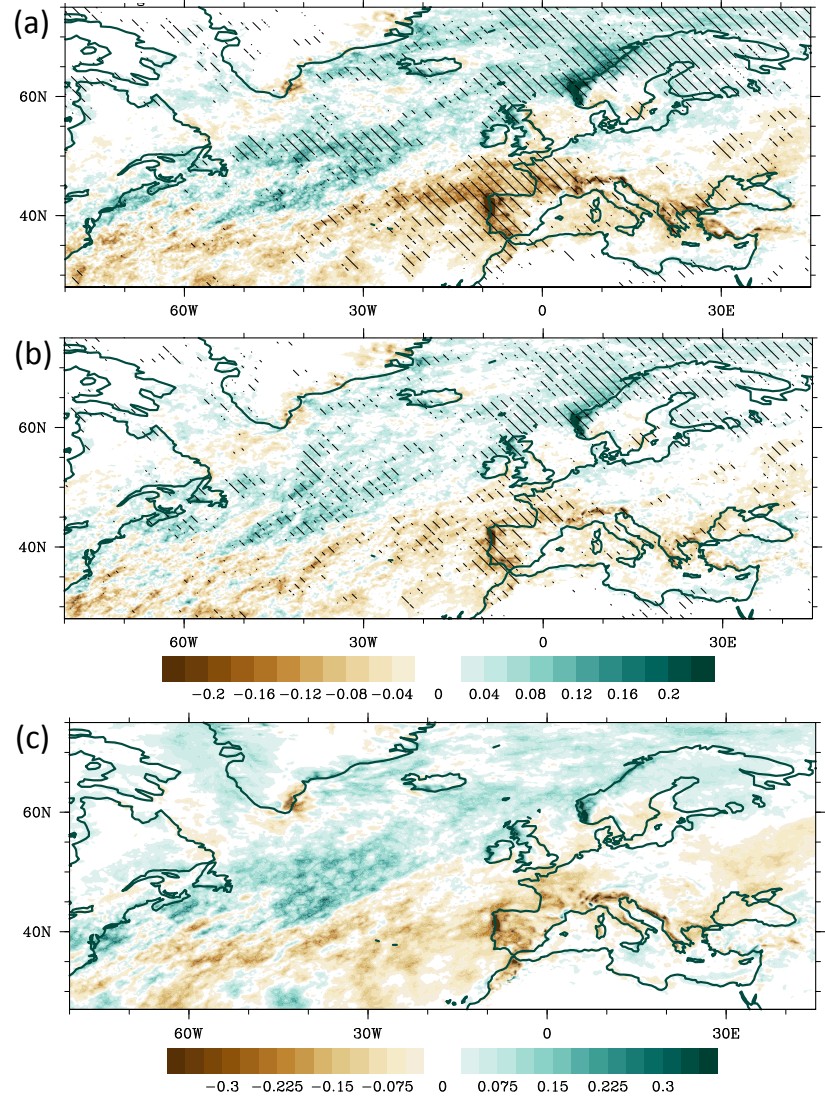

**Figure 6 Difference between +2°C and +1.5°C ensemble experiments for DJF a) 95th percentile of 3hrly precipitation, b) 95th percentile of daily precipitation [mm h-1], c) 10-yr return values in 3hr precipitation CAM5.1.2_0.25. Percentiles and return values are derived from the samples with values larger than 1 mm day-1. Regions in a) and b) are stippled, for differences significant at 10% level.**



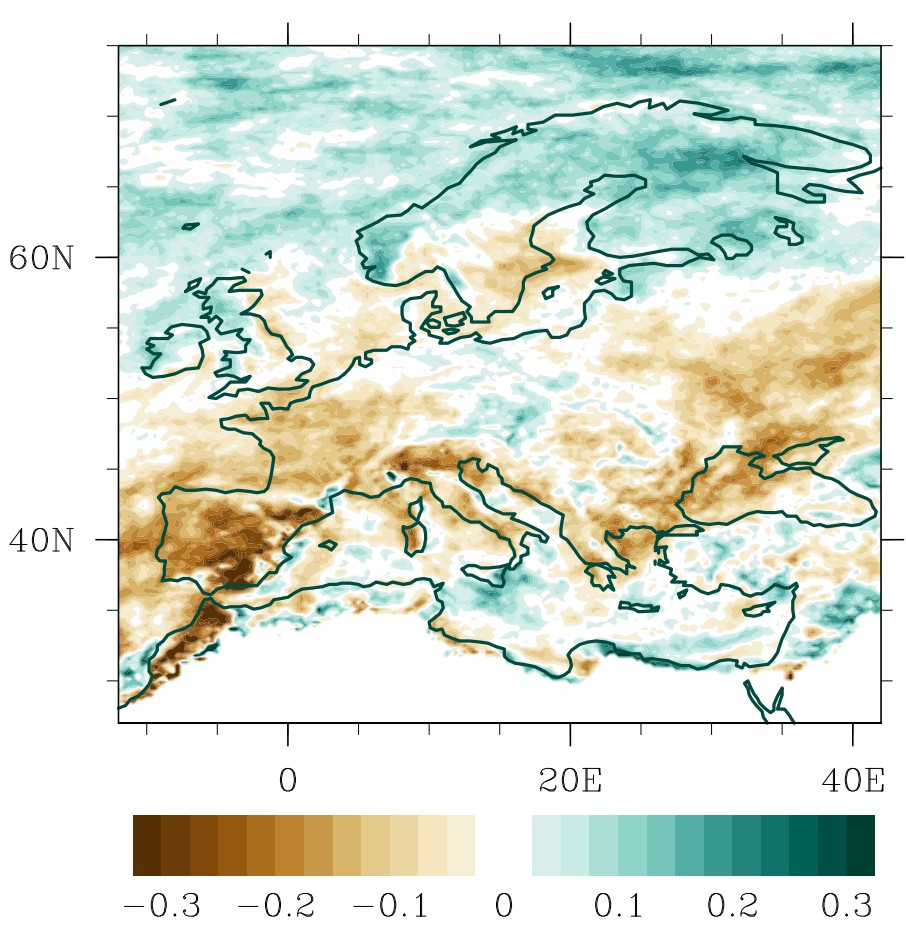

**Figure 7. Fractional difference between +2°C and +1.5°C ensembles (in reference to the +1.5°C experiment) for 10-yr return values of 3hrly precipitation [\*100 %] in CAM5.1.2_0.25. Differences in precipitation were estimated for the values larger than 1 mm day-1.**





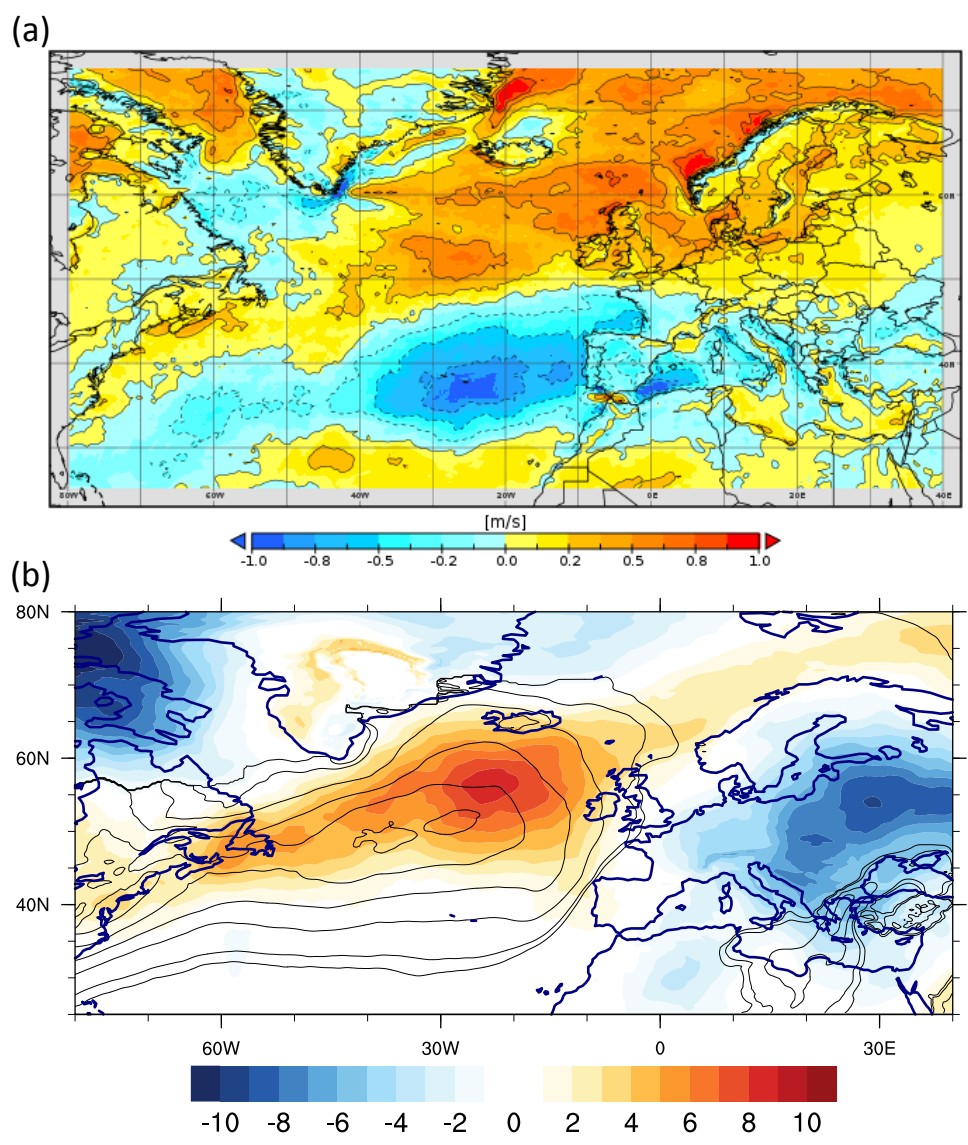

**Figure 8. a) Difference between +2°C and +1.5°C ensemble experiments for DJF) 95$^{th}$ daily wind percentiles, b) DJF 700hPa transient poleward temperature flux in CAM5.1.2_0.25. Contours show the climatology derived for period 1979-2005 (CAM5_0.25) and denote values: 2,4,10, 18, 22, 26 [°C m s$^{-1}$], Values over high orography are masked.**



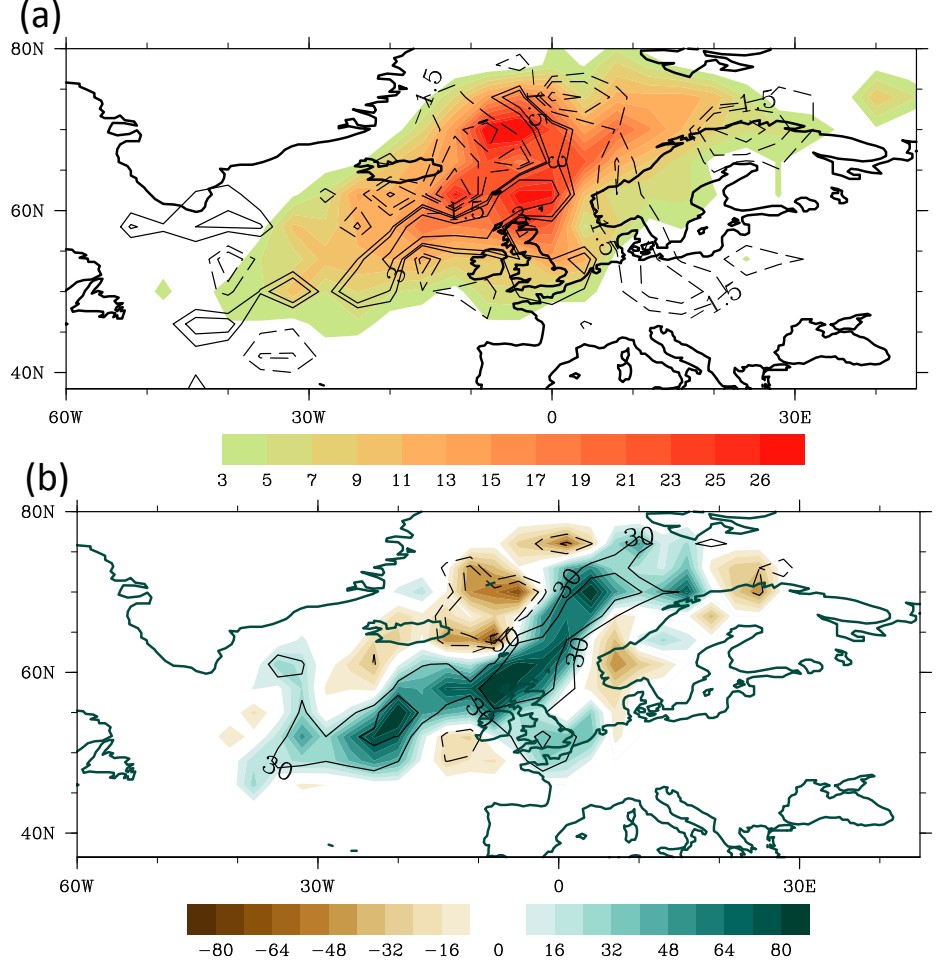

**Figure 9. a) Difference between +2°C minus +1.5°C ensemble experiments (contours) for 50 years sample and mean climatology derived for period 1979-2005 (shaded), for number of 3-h storm**

5     **occurrences accumulated within 4°x 4° grid boxes (number/decade); Climatology derived for period 1979-2005 is shaded.**

**b) Difference between 50 years sample: +2°C minus +1.5°C, estimated for a number of 3-h occurrences with maximum larger than 0.25 [mm h-1], and wind larger than 10m s–1. Maximum values were chosen from 3-h precipitation data on 0.25°deg, which falls into 3°x 3° grid boxes.**

10     **Differences were computed for grid boxes with number over the threshold at least 20/per decade in both experiments.**