# Peer review of "Euro-Atlantic winter storminess and precipitation extremes under 1.5°C versus 2°C warming scenarios"

_Earth System Dynamics, 2017_

## Referee Comment (RC1) · Anonymous Referee #1 · 17 Dec 2017

**1   General Comments**

The manuscript "Euro-Atlantic winter storminess and precipitation extremes under 1.5 °C versus 2 °C warming scenarios" by Barcikowska et al presents comparisons between storminess and precipitation in the 20th century and in the early 22nd century using newly available model simulations from the HAPPI project in different horizontal resolution. They first evaluate model results through comparing results from the model runs on different horizontal resolutions with ERA-Interim (circulation-type variables). Here, they conclude that the 0.25 degree resolution provides the best results, where atmospheric features are presented superior to the lower horizontal resolution model

simulations. In the following, 0.25 degree model precipitation is compared with data from the EOBS and GHCN datasets, where the authors find very good agreement. Afterwards, the authors investigate the differences between the scenarios under 1.5 and 2 °C warming, whereby they present changes in the mean-state of the large-scale atmospheric circulation and precipitation, in daily and sub-daily precipitation and wind extremes, and in storminess for the 0.25 degree run.

Overall, the manuscript deals with an important subject and combines different aspects of how storminess and precipitation changes under 1.5 and 2 °C warming, also with regard to making the model simulation finer. The manuscript clearly conveys this subject, but nevertheless suffers from several major aspects that need improving and/or further clarification, before it is ready to be published.

**2 Specific Comments**

:

1. The HAPPI ensemble consists of several model runs from different modelling centers (not mentioned in the text). I understand that you concentrate on the CAM5-simulations, but give no reason why the other simulations are discarded.

2. Regarding the model resolution: The implications found for the large-scale atmospheric circulation cannot be overstated enough and put the model into the sphere of dynamically downscaled regional models with corrections for the larger scales. I agree with section 3.4 that there is room for some kind of sensitivity study here.

3. I also find it very interesting to see the differences between the present climate and 1.5 °C vs 2 °C warming. Your results suggest (as you wrote) that there

seems to be a threshold in between, that once crossed, exacerbate storminess conditions.

4. Why is ERA Interim the reference for midlatitude atmospheric circulation? How does ERA Interim compare to other reanalyses with regard to circulation?

5. The changes of the SLP gradient are interesting. As you mention the NAO in the beginning of section 3, what are the consequences for the NAO index caused by the increasing SLP gradient difference? You could compute the NAO index and show how it changes as it should be a stationary process centered around 0 in the long-term.

6. This one is very important: The question of statistical significance has not been dealt with properly and currently is rather imprecisely given (sections 2.2, 3.2 and S4).

    S4 and p.12 l18 ("which defies statistical significance"): What you actually show is the distribution of differences, from which you can infer a confidence interval. What you do not get are real implications about statistical significance as written.

    If you want to use bootstrapping to determine statistical significance, the proper way to do it is to generate a null distribution first with a sample size large enough (under H0: zero difference) and then compare your difference with critical values from the null distribution that correspond to your alternative hypothesis (e.g. H1: difference > 0). Using figure S4a (difference in the meridional pressure gradient), I guesstimated a standard deviation of about 2 hPa and generated a normal null distribution for zero differences in R. Comparing your 3 hPa difference with that normal distribution yields significance at least at the 0.9-level. Results may be different for your application as you use a non–parametric approach, depend on the sample size and the test method involved. If the outcome is not significant at any level, there is also room for discussing type-I and type-II errors.

Please repeat your analysis here. Also: please move the detailed description of your method into the method-section without repeating the details later.

Regarding S4: The figure looks very choppy. Either it is showing some kind of histogram sampled for specific blocks of differences (then it should be stated clearly), or it demonstrates undersampling in your bootstrapping approach. Either way, it would be good to redo the bootstrapping with a bigger sample than just 1000. The computation is cheap and very likely results in a better representation of the distribution of differences.

7. The meridional SLP gradient and its differences: Sections 3 and 3.2 write about the SLP gradient, but only refer to figures 4 and 5 showing the respective MSLP plots. I, as a reader, am not able to estimate the gradient and gradient differences from such plots. As you define the gradient in section 2.2, it is very difficult to relate the Azores-Icelandic pressure difference to plots of MSLP or MSLP differences, even though I know about the related atmospheric patterns. Why not just give the gradient as a number somewhere? (also for 1.5 °C and 2 °C scenarios, and the differences).

    Another note to the SLP gradient: It suddenly appears at the end of section 2.2 without prior mentioning. It should be introduced a little earlier along with the other variables (p6 l 32ff) including the reason to do so.

8. For section 3, can you provide spatial statistics, such as the pattern correlation when you describe the resemblence of simulations with observational datasets?

9. Section 3.2, p. 12, l 19 "time-average over 1979-2005": Do you take care of any secular trend, which might be imminent in 26 years of data, but may disappear in a shorter time period? I am asking, because just from looking at the station-based NAO index at https://climatedataguide.ucar.edu/climate-data/hurrell-north-atlantic-oscillation-nao-index-station-based, I see that the average over 1979-2005 is positive, but from 2005-2015 it might average out to 0.

**3  Technical comments**

:

Some passages of the manuscript are not concise (e.g. repetition of methods in the text, when such details belong to the method section). Sometimes the manuscript does not read well.

1. Please check the references. There are references clearly missing, for instance Barcikowska et al, 2017 or Gilleland and Katz, 2014; or misleading like Feser et al., 2014 (did you mean Feser et al., 2015?). There might be more that I have overlooked.

2. Section 3.1 feels a little superfluous and could easily be merged into the method section.

3. p 1 l 32: the British Isles

4. p 4 l 21: Zappa et al. (2014) have shown

5. p 6 l 4: provided by the C20C+ Detection and Attribution Project

6. p 6 l 7: the CAM5-1-1degree run [...] and the CAM5-1-0.25degree run (missing articles)

7. p 6 l 9: add an "and" before 0.3125°x0.234°

8. p 6 l 10: remove the last ")"

9. p 6 l 18: do you need commas in front and after the subclause "using atmosphere-only models?"

10. p 6 l 22: remove comma after offset

11. p 6 l 35: zonal wind; what about meridional winds or wind speeds in general?

12. p 7 l 17: Wilcoxon signed rank test, can you add a reason why you use it?

13. p 7 l 21: a block (seasonal) maximum

14. p 7 l 22: The whole sentence with "Assumptions that our analyzed data..." needs rewriting.

15. p 7 l 26: there is something wrong with (1-1/T)th (you accidently inserted a comma)

16. p 7 l 33-35: please add a reference

17. p 9 l 24: will presumably lead to a

18. p 9 l 37: either from the model bias or from obervational bias

19. p 10 l 4: high-resolution runs provide a more accurate representation

20. p 10 l 20: provided for the years

21. p 10 l 20: internal SST variations being in a different phase

22. p 11 l 18-21: reflect, generally after reductions, CO2 increase

23. Section 3.4: This one reads very well (also applies to section 2.3).

24. Figure 7: What is a fractional difference?

25. Figure 9: The caption talks about 50 years. As far as I understood the manuscript, shouldn't it be less years? Maybe I did not get, where the 50 years sample comes from?

26. Figure S2: There is either something wrong with the figure caption or there is a whole figure missing.

---

## Referee Comment (RC2) · Anonymous Referee #2 · 29 Dec 2017

Title: Euro-Atlantic winter storminess and precipitation extremes under 1.5C versus 2C warming scenarios

Authors: Barcikowska, M. J., Weaver, S. J., Feser, F., Russo, S., Schenk, F., Stone, D. A., and Zahn, M

Ref: Earth Syst. Dynam. Discuss, doi:10.5194/esd-2017-106, in review, 2017.

Recommendation: major revisions needed

General Comments: The authors have submitted a very interesting study analyzing the implications of climate change to wind and precipitation over Europe using a small

ensemble of high resolution climate model simulations. The objective is to quantify the differences between a 1,5C and a 2C scenario, focusing on extreme precipitation and extreme winds on the regional scale. However, several aspects of the study are not well explained or needed to be revised before the publication. In particular, I find it hard to understand the differences between the 1,5C and 2C scenarios as externally forced – this could as well be internal variability or some issue with the set up of the time slice experiments. The comments below should help the authors revise the manuscript and re-submit the paper to ESD.

Main Comment: A: Based on their analysis of the high resolution time slide experiments, the authors state that most of the changes due to anthropogenic forcing up to the 2C level only start occurring after the 1,5C threshold has been surpassed, and claim this may be due to non-linearities in the climate system (page 17, around line 30). While this may well be true to some point, I find it very hard to believe that while the climate change signal e.g. DJF MSLP and 850 hPa winds up to a 1,5C warmer world is practically zero (cf. Fig. S1a), we become a very strong response when we add the additional 0,5C (cf. Fig. S1b). The same could be argued for the other fields. In my view, such a strong difference for such a small increment of external forcing could only come from (a) internal variability (for example, storminess is at a low level of its decadal variability in the period chosen for the 1,5C experiment, while it is on a high level on the period chosen for 2C) and/or (b) there are some issue in the set up of the high resolution simulations which have lead to these differences, and not the small change in the forcing. This does not turn the results per themselves wrong, but means that the authors may be misinterpreting (or at least over-interpreting) their results. In can be that the problem is related with the SST set up, as the authors shortly discuss in section 2.1. For example, how does the climate change signal look like for the (hopefully transient) lower resolution GCM simulations? How does the climate change signal look like for the single ensemble members? This would be important to analyse in detail to identify if the changes in precipitation and wind are continuous over time (plus natural decadal climate variability) or if indeed some strong "non-linear" effects occur.

It is very improbable that the AMOC will collapse in a two degree warmer world. What might be important here is the thermodynamical effect (Clausius-Clapeyron), primarily for precipitation intensity. This is very important to rule out that the obtained results are not simply caused by some issue with the set up of the model simulations and clearly relate the changes to the increases of anthropogenic forcing.

Minor Comments: (Note: I have focussed primarily on the introduction, as the other parts may change during revision)

**1 Page 2, line 34: regarding the role of cyclone clustering for the wind and flood impacts, the authors could refer for example to Priestley et al. (2017), which deals specifically with this topic.**

**2: Page 3, around line 10: The "poleward" shift of the storm tracks due to enhanced anthropogenic forcing is correct on zonal average, but this is not always true for Europe: while this is clear for the summer half year, the results for the winter, particularly DJF (the focus here), are different – see your own text in page 4, line 10. Please check e.g., the discussion in Zappa et al (2015), and also their Fig.1.**

**3: Page 4, line 12: The results by Zappa et al. (2013b) do not contradict the results described in many of the papers mentioned in the previous sentence. For example, the results from Bengtsson et al. (2006), Pinto et al. (2009) or Zappa et al. (2013b) only show small differences in detail regarding the intensification of cyclone activity over the British Isles and the strong decrease in the Mediterranean. Thus, "however" is not the correct word here, maybe "Morevoer" would be more appropriate.**

**4: Page 4/5: The authors describe their reasoning as if it would be the very first time that high-resolution global climate modelling is performed at ∼25km resolution. This is not correct – for example, there is a whole EU project regarding high resolution modelling to study impacts of climate change for Europe (PRIMAVERA, https://www.primavera-h2020.eu/about/objectives/). It is strange not to mention this at all in the introduction, nor any of the related papers (e.g. Schiemann et al., 2017).**

The MIROC and MRI groups has also been working high-resolution climate models for several years (e.g. Murakami et al., 2011), which is not really discussed– tough Kitoh and Endo (2016) is shortly mentioned. Please note that another possibility would be to use RCMs as ESMs (e.g. Sein et al., 2015). Please enhance.

**5: References: many issues, both in the reference list and in the text. For example, Zappa et al (2012) is surely one of the two Zappa et al. (2013) papers, Feser et al (2014) is probably Feser et al. (2015). Pinto et al (2009) is three times in the reference list, and several references mentioned in the text are not included (e.g. Barcikowska et al. 2017) are not included in the Reference List. Please enhance.**

**6: Figures: it would be very good to had labels to the isolines in the figures, particularly in Figures 8 and 9. It is not enough to mention the values in the captions. Please enhance.**

Suggested additional References

Murakami et al (2011) JCLIM, https://doi.org/10.1175/2010JCLI3723.1

Priestley et al (2017) Weather, https://doi.org/10.1002/wea.3025

Schiemann et al. (2017) JCLIM, https://doi.org/10.1175/JCLI-D-16-0100.1

Sein et al. (2015) JAMES, https://doi.org/10.1002/2014MS000357

Zappa et al. (2015) JCLIM, https://doi.org/10.1175/JCLI-D-14-00823.1
* * *

---

## Author Comment (AC1) · 9 Feb 2018

We thank referee #1 for his constructive review. We will update our analysis with new present climate runs, which were not available previously. Unlike the historical 1979-2005 simulation, the new runs now follow the HAPPI experiment protocol, i.e. they constitute an ensemble of decadal runs in 2006-2015. Therefore they are more suitable for the analysis investigating changes between the present and future climate. With the updated/improved design of the analysis, some of the reviewer's comments may not apply to the current version of the paper. We will provide explanations below.

Anonymous Referee #1

[Figure]

1 General Comments The manuscript "Euro-Atlantic winter storminess and precipitation extremes under 1.5 _C versus 2 _C warming scenarios" by Barcikowska et al presents comparisons between storminess and precipitation in the 20th century and in the early 22nd century using newly available model simulations from the HAPPI project in different horizontal resolution. They first evaluate model results through comparing results from the model runs on different horizontal resolutions with ERA-Interim (circulation-type variables). Here, they conclude that the 0.25 degree resolution provides the best results, where atmospheric features are presented superior to the lower horizontal resolution model simulations. In the following, 0.25 degree model precipitation is compared with data from the EOBS and GHCN datasets, where the authors find very good agreement. Afterwards, the authors investigate the differences between the scenarios under 1.5 and 2 _C warming, whereby they present changes in the mean-state of the large-scale atmospheric circulation and precipitation, in daily and sub-daily precipitation and wind extremes, and in storminess for the 0.25 degree run. Overall, the manuscript deals with an important subject and combines different aspects of how storminess and precipitation changes under 1.5 and 2 _C warming, also with regard to making the model simulation finer. The manuscript clearly conveys this subject, but nevertheless suffers from several major aspects that need improving and/or further clarification, before it is ready to be published.

1. The HAPPI ensemble consists of several model runs from different modeling centers (not mentioned in the text). I understand that you concentrate on the CAM5-simulations, but give no reason why the other simulations are discarded.

AU: In this study, we only focus on the same model version run at different resolutions. This allows us to investigate the impacts of a very high model resolution on the representation of large-scale and regional features in comparison to a coarser resolution. Additionally CAM5 1.2-0.25 provides unprecedented opportunity to investigate extremes on subdaily time scales. We stated this in the introduction, but will make it clearer in the revised version.

2. Regarding the model resolution: The implications found for the large-scale atmospheric circulation cannot be overstated enough and put the model into the sphere of dynamically downscaled regional models with corrections for the larger scales. I agree with section 3.4 that there is room for some kind of sensitivity study here.

AU: We agree but think that such sensitivity studies should be part of a separate study. We nevertheless motivate such studies in our manuscript.

3. I also find it very interesting to see the differences between the present climate and 1.5 _C vs 2 _C warming. Your results suggest (as you wrote) that there seems to be a threshold in between, that once crossed, exacerbate storminess conditions.

AU: Yes, one interpretation could be associated with the threshold. Other interpretation, with an additional analysis including new present climate simulations, suggests that the difference might be due to the asymmetry in aerosol forcing between the present and future climate. It is a very important point and we elaborate on that in the revised version. We support our statements, by updating the analysis with new simulations of the present climate. These simulations follow the HAPPI protocol (unlike the previous ones) and are more relevant to address this issue.

4. Why is ERA Interim the reference for midlatitude atmospheric circulation? How does ERA Interim compare to other reanalyses with regard to circulation?

AU: We use ERA interim, because its spatial resolution is comparable to the high-resolution CAM5 model simulations. ERA interim has considerably higher model resolution (80 km at 60 vertical levels) than other reanalysis products. Hence it is able to resolve sharper spatial gradients than e.g. NCEP/NCAR Reanalysis. Hodges et al. 2011 compares reanalyses for extratropical cyclones and shows that the newer reanalyzes (especially ERA-Interim and NCEP-CFSR) agree (both in terms of numbers and locations) much better than the older ones (JRA-25) for both hemispheres and that intensities are higher. As our purpose is to validate the model, ERA interim serves the purpose better coarse reanalysis products like NCEP/NCAR.

Hodges, K.I., R.W. Lee, and L. Bengtsson, 2011: A Comparison of Extratropical Cyclones in Recent Reanalyses ERA-Interim, NASA MERRA, NCEP CFSR, and JRA-25. J. Climate, 24, 4888–4906, https://doi.org/10.1175/2011JCLI4097.1

5. The changes of the SLP gradient are interesting. As you mention the NAO in the beginning of section 3, what are the consequences for the NAO index caused by the increasing SLP gradient difference? You could compute the NAO index and show how it changes as it should be a stationary process centered around 0 in the long-term.

AU: Because of the strengthening of the SLP gradient, NAO index will likely have a positive tendency, when compared +2C future with the +1.5C/present climate experiments. However under the each stabilization scenario the spatial pattern/definition of NAO is different. Hence in this experimental set up NAO could be rather investigated separately for each of the three ensembles. Therefore more thorough analysis could be done only on NAO, but in a separate study. Otherwise the transient simulation would be more relevant for addressing the reviewer's question.

6. This one is very important: The question of statistical significance has not been dealt with properly and currently is rather imprecisely given (sections 2.2, 3.2 and S4). S4 and p.12 l18 ("which defies statistical significance"): What you actually show is the distribution of differences, from which you can infer a confidence interval. What you do not get are real implications about statistical significance as written.

Please repeat your analysis here. Also: please move the detailed description of your method into the method-section without repeating the details later. Regarding S4: The figure looks very choppy. Either it is showing some kind of histogram sampled for specific blocks of differences (then it should be stated clearly), or it demonstrates undersampling in your bootstrapping approach. Either way, it would be good to redo the bootstrapping with a bigger sample than just 1000. The computation is cheap and very likely results in a better representation of the distribution of differences.

AU: This is indeed an important point. This analysis would be certainly very helpful

for understating the implications for statistical significance, at the absence of the simulations, which are design to cancel out the effect of the internal variability. Instead, we will update the analysis with new simulations and discussion. These are tailored specifically to remove the effects of different phase of internal variability, questioning the statistical significance of the derived results.

7. The meridional SLP gradient and its differences: Sections 3 and 3.2 write about the SLP gradient, but only refer to figures 4 and 5 showing the respective MSLP plots. I, as a reader, am not able to estimate the gradient and gradient differences from such plots. As you define the gradient in section 2.2, it is very difficult to relate the Azores-Icelandic pressure difference to plots of MSLP or MSLP differences, even though I know about the related atmospheric patterns. Why not just give the gradient as a number somewhere? (also for 1.5 _C and 2 _C scenarios, and the differences).

Another note to the SLP gradient: It suddenly appears at the end of section 2.2 without prior mentioning. It should be introduced a little earlier along with the other variables (p6 l 32ff) including the reason to do so.

AU: We will add the information and improve the structure of the presented results.

8. For section 3, can you provide spatial statistics, such as the pattern correlation when you describe the resemblence of simulations with observational datasets?

AU: Yes, we will update the analysis with the pattern correlation.

9. Section 3.2, p. 12, l 19 "time-average over 1979-2005": Do you take care of any secular trend, which might be imminent in 26 years of data, but may disappear in a shorter time period?

AU: That is a valid point, which we would certainly take into consideration. However, in the updated version of the analysis we no longer use the 1979-2005 run. Instead we use new runs of present climate. These simulations are more suitable as they are set to follow the HAPPI protocol. We will update the revised version of the paper accordingly.

[Figure]

3. Technical comments:

Some passages of the manuscript are not concise (e.g. repetition of methods in the text, when such details belong to the method section). Sometimes the manuscript does not read well.

AU: We will edit our text accordingly to improve clarity and readability of the manuscript.

1. Please check the references. There are references clearly missing, for instance Barcikowska et al, 2017 or Gilleland and Katz, 2014; or misleading like Feser et al., 2014 (did you mean Feser et al., 2015?). There might be more that I have overlooked.

AU: We will add and correct the references in the revised version.

2. Section 3.1 feels a little superfluous and could easily be merged into the method section.

AU: The section will be shorter and more concise in our revised version, as it excludes the analysis of internal variability impacts.

AU: We will apply all corrections, as suggested below.

3. p 1 l 32: the British Isles 4. p 4 l 21: Zappa et al. (2014) have shown 5. p 6 l 4: provided by the C20C+ Detection and Attribution Project 6. p 6 l 7: the CAM5-1-1degree run [...] and the CAM5-1-0.25degree run (missing articles) 7. p 6 l 9: add an "and" before 0.3125_x0.234_ 8. p 6 l 10: remove the last ")" 9. p 6 l 18: do you need commas in front and after the subclause "using atmosphereonly models?" 10. p 6 l 22: remove comma after offset 11. p 6 l 35: zonal wind; what about meridional winds or wind speeds in general? 12. p 7 l 17: Wilcoxon signed rank test, can you add a reason why you use it? 13. p 7 l 21: a block (seasonal) maximum 14. p 7 l 22: The whole sentence with "Assumptions that our analyzed data..." needs rewriting. 15. p 7 l 26: there is something wrong with (1-1/T)th (you accidently inserted a comma) 16. p 7 l 33-35: please add a reference AU: We will apply the correction 17. p 9 l 24: will presumably lead to a 18. p 9 l 37: either from the model bias or from obervational bias

19. p 10 l 4: high-resolution runs provide a more accurate representation 20. p 10 l 20: provided for the years 21. p 10 l 20: internal SST variations being in a different phase 22. p 11 l 18-21: reflect, generally after reductions, CO2 increase 23. Section 3.4: This one reads very well (also applies to section 2.3).

24. Figure 7: What is a fractional difference?

AU: We will provide a definition of a fractional difference, which is simply percentage of change in a relation to the mean in a reference period.

25. Figure 9: The caption talks about 50 years. As far as I understood the manuscript, shouldn't it be less years? Maybe I did not get, where the 50 years sample comes from?

AU: 50yrs sample constitute a 5-member ensemble of decadal runs. We will clarify it in the revised manuscript.

26. Figure S2: There is either something wrong with the figure caption or there is a whole figure missing.

AU: Yes, we will correct the caption and also update the figure with an analysis using new present climate simulations.

---

## Author Comment (AC2) · 9 Feb 2018

We thank referee #2 for the constructive review. We will update our analysis with new present climate runs, which were not available previously. Unlike the historical 1979-2005 simulation, the new runs now follow the HAPPI experiment protocol, i.e. they constitute an ensemble of decadal runs in 2006-2015. Therefore they are more suitable for the analysis investigating changes between the present and future climate. This approach is also more suitable to address the main reviewers concerns on the 'non-linear effects of the derived climate change, shown in the study. We will provide explanations below.

[Figure]

A: Based on their analysis of the high resolution time slide experiments, the authors state that most of the changes due to anthropogenic forcing up to the 2C level only start occurring after the 1,5C threshold has been surpassed, and claim this may be due to non-linearities in the climate system (page 17, around line 30). While this may well be true to some point, I find it very hard to believe that while the climate change signal e.g. DJF MSLP and 850 hPa winds up to a 1,5C warmer world is practically zero (cf. Fig. S1a), we become a very strong response when we add the additional 0,5C (cf. Fig. S1b). The same could be argued for the other fields.

In my view, such a strong difference for such a small increment of external forcing could only come from (a) internal variability (for example, storminess is at a low level of its decadal variability in the period chosen for the 1,5C experiment, while it is on a high level on the period chosen for 2C) and/or (b) there are some issue in the set up of the high resolution simulations which have lead to these differences, and not the small change in the forcing. This does not turn the results per themselves wrong, but means that the authors may be misinterpreting (or at least over-interpreting) their results. In can be that the problem is related with the SST set up, as the authors shortly discuss in section 2.1. For example, how does the climate change signal look like for the (hopefully transient) lower resolution GCM simulations? How does the climate change signal look like for the single ensemble members? This would be important to analyse in detail to identify if the changes in precipitation and wind are continuous over time (plus natural decadal climate variability) or if indeed some strong "non-linear" effects occur. It is very improbable that the AMOC will collapse in a two degree warmer world. What might be important here is the thermodynamical effect (Clausius-Clapeyron), primarily for precipitation intensity. This is very important to rule out that the obtained results are not simply caused by some issue with the set up of the model simulations and clearly relate the changes to the increases of anthropogenic forcing.

AU: Thank you for the comment. Indeed, the "non-linear" effects of the derived changes, when comparing first half a degree warming (+1.5C minus present) with the
additional half a degree warming (+2C minus +1.5C), are likely associated with the simulations set up. i.e. an asymmetry in the aerosol forcing between the present and future climate scenarios. The changes associated with warming at the 1.5°C level stem from an interplay of a number of forcings, including strong aerosol reductions, while an additional half a degree warming is solely a consequence of $CO_2$ increases and ocean warming. We mentioned this in the manuscript, but we will enhance this message in the revised version. This will be also supported with an updated analysis, which includes new present climate simulations. These simulations follow the HAPPI protocol (unlike the previous ones) and are more relevant to address the nonlinearity issue.

Minor comments:

**1 Page 2, line 34: regarding the role of cyclone clustering for the wind and flood impacts, the authors could refer for example to Priestley et al. (2017), which deals specifically with this topic. AU: Thank you, we can apply this suggestion.**

**2: Page 3, around line 10: The "poleward" shift of the storm tracks due to enhanced anthropogenic forcing is correct on zonal average, but this is not always true for Europe: while this is clear for the summer half year, the results for the winter, particularly DJF (the focus here), are different – see your own text in page 4, line 10. Please check e.g., the discussion in Zappa et al (2015), and also their Fig.1.**

AU: Yes, we will take this comment into consideration while correcting the manuscript.

**3: Page 4, line 12: The results by Zappa et al. (2013b) do not contradict the results described in many of the papers mentioned in the previous sentence. For example, the results from Bengtsson et al. (2006), Pinto et al. (2009) or Zappa et al. (2013b) only show small differences in detail regarding the intensification of cyclone activity over the British Isles and the strong decrease in the Mediterranean. Thus, "however" is not the correct word here, maybe "Morevoer" would be more appropriate.**

AU: Thank you for the correction.

**4: Page 4/5: The authors describe their reasoning as if it would be the very first time that high-resolution global climate modelling is performed at _25km resolution. This is not correct – for example, there is a whole EU project regarding high resolution modelling to study impacts of climate change for Europe (PRIMAVERA, https://www.primavera-h2020.eu/about/objectives/). It is strange not to mention this at all in the introduction, nor any of the related papers (e.g. Schiemann et al., 2017). The MIROC and MRI groups has also been working high-resolution climate models for several years (e.g. Murakami et al., 2011), which is not really discussed– tough Kitoh and Endo (2016) is shortly mentioned. Please note that another possibility would be to use RCMs as ESMs (e.g. Sein et al., 2015). Please enhance.**

AU: Thank you for the correction, we will certainly clarify and enhance the Introduction.

**5: References: many issues, both in the reference list and in the text. For example, Zappa et al (2012) is surely one of the two Zappa et al. (2013) papers, Feser et al (2014) is probably Feser et al. (2015). Pinto et al (2009) is three times in the reference list, and several references mentioned in the text are not included (e.g. Barcikowska et al. 2017) are not included in the Reference List. Please enhance.**

AU: We will check again all references.

**6: Figures: it would be very good to had labels to the isolines in the figures, particularly in Figures 8 and 9. It is not enough to mention the values in the captions. Please enhance. AU: We will also include the suggested additional references.**

---

## Author Response (AR1)

We would like to thank the reviewer for careful reading of this manuscript and for the constructive comments. We have responded to all comments below, added explanations where necessary, and applied most of the suggested changes or corrections.

We have updated our analysis with new present climate runs, which were not available previously. Unlike the historical 1979-2005 simulation, the new runs now follow the HAPPI experiment protocol, i.e. they constitute an ensemble of decadal runs in 2006-2015. Therefore they are more suitable for the analysis investigating changes between the present and future climate. With the updated/improved design of the analysis, some of the reviewer's comments may not apply to the current version of the paper. We will provide more explanations below.

Anonymous Referee #1

Received and published: 17 December 2017

1 General Comments

The manuscript "Euro-Atlantic winter storminess and precipitation extremes under 1.5 \_C versus 2 \_C warming scenarios" by Barcikowska et al presents comparisons between storminess and precipitation in the 20th century and in the early 22nd century using newly available model simulations from the HAPPI project in different horizontal resolution. They first evaluate model results through comparing results from the model runs on different horizontal resolutions with ERA-Interim (circulation-type variables). Here, they conclude that the 0.25 degree resolution provides the best results, where atmospheric features are presented superior to the lower horizontal resolution model simulations. In the following, 0.25 degree model precipitation is compared with data from the EOBS and GHCN datasets, where the authors find very good agreement.

Afterwards, the authors investigate the differences between the scenarios under 1.5 and 2 \_C warming, whereby they present changes in the mean-state of the large-scale atmospheric circulation and precipitation, in daily and sub-daily precipitation and wind extremes, and in storminess for the 0.25 degree run. Overall, the manuscript deals with an important subject and combines different aspects of how storminess and precipitation changes under 1.5 and 2 \_C warming, also with regard to making the model simulation finer. The manuscript clearly conveys this subject, but nevertheless suffers from several major aspects that need improving and/or further clarification, before it is ready to be published.

1. The HAPPI ensemble consists of several model runs from different modeling centers (not mentioned in the text). I understand that you concentrate on the CAM5-simulations, but give no reason why the other simulations are discarded.

AU: In this study, we only focus on the same model version run at different resolutions. This allows us to investigate the impacts of a very high model resolution on the representation of large-scale and regional features in comparison to a coarser resolution. Additionally CAM5 1.2-0.25 provides unprecedented opportunity to investigate extremes on subdaily time scales. We stated this in the introduction, but will make it clearer in the revised version. Also we don't feel obliged to list all the modeling centers participating in the HAPPI projects. In similar way, studies employing particular CMIP models don't usually list all of the other centers/models, contributing to the CMIP models community. (page 5, line 30-35)

2. Regarding the model resolution: The implications found for the large-scale atmospheric circulation cannot be overstated enough and put the model into the sphere of dynamically downscaled regional models with corrections for the larger scales. I agree with section 3.4 that there is room for some kind of sensitivity study here.

AU: We agree but think that such sensitivity studies should be part of a separate study. We nevertheless motivate such studies in our manuscript.

3. I also find it very interesting to see the differences between the present climate and 1.5 \_C vs 2 \_C warming. Your results suggest (as you wrote) that there seems to be a threshold in between, that once crossed, exacerbate storminess conditions.

AU: Yes, one interpretation could be associated with the threshold. Other interpretation, with an additional analysis including new present climate simulations, suggests that the difference might be due to the asymmetry in aerosol forcing between the present and future climate. It is a very important point and we elaborate on that in the revised version. We support our statements, by updating the analysis with new simulations of the present climate. These simulations follow the HAPPI protocol (unlike the previous ones) and are more relevant to address this issue. (page 7, lines 3-9; 27-29; page 11, lines: 1-9, page 12, lines: 10-21; page 17, lines 25-30)

4. Why is ERA Interim the reference for midlatitude atmospheric circulation? How does ERA Interim compare to other reanalyses with regard to circulation?

AU: We use ERA interim, because its spatial resolution is comparable to the high-resolution CAM5 model simulations. ERA interim has considerably higher model resolution (80 km at 60 vertical levels) than other reanalysis products. Hence it is able to resolve sharper spatial gradients than e.g. NCEP/NCAR Reanalysis. Hodges et al. 2011 compares reanalyses for extratropical cyclones and shows that the newer reanalyzes (especially ERA-Interim and NCEP-CFSR) agree (both in terms of numbers and locations) much better than the older ones (JRA-25) for both hemispheres and that intensities are higher. As our purpose is to validate the model, ERA interim serves the purpose better than the coarse reanalysis products like NCEP/NCAR.

We updated the Figure 1a and relevant text in the manuscript, by adding the comparison of the ERA-I and NCEP-DOE data sets. It shows that differences between the observations are much smaller than those derived between observations and CAM5. See changes in 2.1 section: **page 7 line,14-16; page 9 line: 13-22.**

The figures attached below differentiates top) ERA-I and CFSR, bottom) NCEP and ERA-I, showing that for the North Atlantic-European region, the largest observational uncertainty is located over Greenland. Observational uncertainty is much smaller, compared to the differences between the observations and the model.

We also added reference (page 9, line 16):

Hodges, K.I., R.W. Lee, and L. Bengtsson, 2011: A Comparison of Extratropical Cyclones in Recent Reanalyses ERA-Interim, NASA MERRA, NCEP CFSR, and JRA-25. *J. Climate*, 24, 4888–4906, https://doi.org/10.1175/2011JCLI4097.1

Figure. Time-mean average of the DJF sea level pressure [hPa] over the period 1979-2005, regridded to the 2.5°x2.5° horizontal grid for top) ERA-Interim (shaded) and differences relative to CFSR (at original resolution T382, ~0.34° resolution at equator); bottom) NCEP/DOE 2 (~2.5 ° original resolution, shaded). Contours show differences relative to ERA-Interim (ERA-I, ~0.75° latlon original resolution)

5. The changes of the SLP gradient are interesting. As you mention the NAO in the beginning of section 3, what are the consequences for the NAO index caused by the increasing SLP gradient difference? You could compute the NAO index and show how it changes as it should be a stationary process centered around 0 in the long-term.

AU: Because of the strengthening of the SLP gradient, the NAO index will likely have a positive tendency, when compared to a +2C future with the +1.5C/present climate experiments. However under each stabilization scenario, the spatial pattern/definition of the NAO would be different. Hence in this experimental setup, the NAO could be rather investigated separately for each of the three ensembles. Therefore a more thorough analysis could be done only on the NAO, but in a separate study and based on transient simulations rather than time slice experiments.

6. This one is very important: The question of statistical significance has not been dealt with properly and currently is rather imprecisely given (sections 2.2, 3.2 and S4). S4 and p.12 118 ("which defies statistical significance"): What you actually show is the distribution of differences, from which you can infer a confidence interval. What you do not get are real implications about statistical significance as written.

Please repeat your analysis here. Also: please move the detailed description of your method into the method-section without repeating the details later. Regarding S4: The figure looks very choppy. Either it is showing some kind of histogram sampled for specific blocks of differences (then it should be stated clearly), or it demonstrates undersampling in your bootstrapping approach. Either way, it would be good to redo the bootstrapping with a bigger sample than just 1000. The computation is cheap and very likely results in a better representation of the distribution of differences.

AU: This is indeed an important point. We would certainly repeat this analysis, at the absence of present climate simulations following the HAPPI protocol. Instead, we will update the analysis with new simulations and discussion. These are tailored specifically to remove the effects of different phase of internal variability, which was questioning the statistical significance of the derived results.

In the new version of section 3.1, we were now able to use new ensemble simulations of the present climate. These runs represent the 2006-2015 period, instead the previously used simulation of the 1979-2005 period. The new runs follow closely HAPPI protocol, and thus facilitate more direct interpretation of the results. The new present climate runs, as well as each of the future warming, i.e. +1.5C and +2C experiment includes internal climate SST variations (e.g. ENSO) during the same decadal period, i.e. 2006-2015. Therefore it is expected that the impacts of internal variations will be canceled out while discriminating between all three experiments. In this context we found the bootstrapping approach to be redundant and we removed this part of analysis (and associated figure S4).

Please note that for the analysis of the climatological features of the model and the impact of the resolution (e.g. section 3, and 3.3), we used historical runs for 1979-2005.

It is worth noting, that the asymmetry in forcing, when comparing the present and future climate experiments, remains a factor complicating the interpretation of the results. The changes associated with warming at the  $1.5^{\circ}$ C level stem from an interplay of a number of forcings, including strong aerosol reductions, while an additional half a degree warming is solely a consequence of further CO2 increase and ocean warming (**page 17, line 25-30**). Please see more explanation: page 11, lines: 1-10, page 12, line: 15-20.

7. The meridional SLP gradient and its differences: Sections 3 and 3.2 write about the SLP gradient, but only refer to figures 4 and 5 showing the respective MSLP plots. I, as a reader, am not able to estimate the gradient and gradient differences from such plots. As you define the gradient in section 2.2, it is very difficult to relate the Azores-Icelandic pressure difference to plots of MSLP or MSLP differences, even though I know about the related atmospheric patterns. Why not just give the gradient as a number somewhere? (also for 1.5 \_C and 2 \_C scenarios, and the differences).

AU: The estimated changes in the SLP gradient are consistent with the derived changes in the large-scale circulation. We included the numbers and the relevant discussion in the text: section 3.1 page 11, line: 15-18, page12: 12-14, 15-20.

We also underlined the fact that the estimations of the differences between the  $\pm 1.5$ C and  $\pm 2$ °C scenario are most pronounced and that we will focus mostly on the differences derived between the  $\pm 1.5$ C and  $\pm 2$ °C scenario. The explanation is provided in the page **12**, **line 15-21**.

Additionally, contours (labeled) of SLP in Figure 4a and 5a show the time-average SLP in the present climate, while the differences between the experiments are shaded. This facilitates the interpretation of the future SLP changes in the context of the mean ambient flow.

Another note to the SLP gradient: It suddenly appears at the end of section 2.2 without prior mentioning. It should be introduced a little earlier along with the other variables (p6 1 32ff) including the reason to do so.

AU: Yes, we provided and expanded the information on the SLP gradient in the earlier part of the methods section (page 7, line 37 - page 8 line 1-4).

8. For section 3, can you provide spatial statistics, such as the pattern correlation when you describe the resemblence of simulations with observational datasets?

AU: Yes, we have updated the analysis with the pattern correlation.

The observed average SLP patterns, derived from ERA-I and NCEP/DOE 2 ( $\sim 2.5^{\circ}$ ) share a correlation (uncentered pattern correlation) of 0.98. The differences between the ERA-I and CAM5 are larger, than the observational differences.

The magnitude and the pattern in ERA-I correlate best with the one simulated at similar horizontal resolution (CAM5\_1), returning number of 0.96. Correlations with the remaining two are slightly smaller, i.e 0.95 for CAM5\_2 and 0.94 for CAM5\_0.25. Please see in text (Section 3, page 9: lines 26-29).

9. Section 3.2, p. 12, 1 19 "time-average over 1979-2005": Do you take care of any secular trend, which might be imminent in 26 years of data, but may disappear in a shorter time period?

AU: That is a valid point, which we would certainly take into consideration. However, in the updated version of the analysis in this section, we no longer use the 1979-2005 run. Instead, we use new runs of the present climate. These simulations are more suitable as they are set to follow the HAPPI protocol. We will update the revised version of the paper accordingly.

In this version we are using decadal runs (2006-2015). The impact of secular trend in these runs is rather negligible. (methods section: **page 7**, **line 28-29**)

3. Technical comments:

Some passages of the manuscript are not concise (e.g. repetition of methods in the text, when such details belong to the method section). Sometimes the manuscript does not read well.

AU: We worked on readability of the manuscript, i.e. section 3 is shortened and clarified. We merged section 3.1 and 3.2. Some technical information is moved to the method section (page 7, line 28-29, page 7, line 36-37; page 8, line 1-3). But we insist on repeating some technical facts in the introduction of section 3.1, instead mentioning them only in the methods, to facilitate the interpretation of the following results. (page 10, line 35 - page 11, line 10).

1. Please check the references. There are references clearly missing, for instance Barcikowska et

al, 2017 or Gilleland and Katz, 2014; or misleading like Feser et al., 2014 (did you mean Feser et al., 2015?). There might be more that I have overlooked.

AU: We corrected the references in the revised version.

2. Section 3.1 feels a little superfluous and could easily be merged into the method section.

AU: The section will be shorter and more concise in our revised version, as it excludes the analysis of internal variability impacts.

As suggested, we merged section 3.1 and old version of section 3.2. As mentioned above, some part of information, e.g. description of the anlaysis of the large –scale circulation is moved to the method section. However we left the information, which facilitates the interpretation of the following results. (page 10, line 35-37; page 11: line:1-9)

AU: We have applied corrections, as suggested below:

3. p 1 1 32: the British Isles  $\rightarrow$  corrected (page 4, line 13)

- 4. p 4 l 21: Zappa et al. (2014) have shown:  $\rightarrow$  corrected (page 4, line 22)
- 5. p 6 1 4: provided by the C20C+ Detection and Attribution Project  $\rightarrow$  corrected
- 6. p 6 1 7: the CAM5-1-1degree run [...] and the CAM5-1-0.25degree run (missing articles)  $\rightarrow$  corrected (page 6, line 13-15)

7. p 619: add an "and" before  $0.3125_{x}0.234 \rightarrow$  corrected (page 6, line 16)

8. p 6 l 10: remove the last ")"  $\rightarrow$  corrected (page 6, line 18)

9. p 6 1 18: do you need commas in front and after the subclause "using atmosphereonly

models?"  $\rightarrow$  we think that commas will improve clarity of the sentence (page 6, line 25)

10. p 6 l 22: remove comma after offset  $\rightarrow$  corrected (page 6, line 29)

11. p 6 1 35: zonal wind; what about meridional winds or wind speeds in general?  $\rightarrow$  edited as "winds" (page 7, line 11)

12. p 7 l 17: Wilcoxon signed rank test, can you add a reason why you use it?  $\rightarrow$  added in page 7: line 31-32

13. p 7 l 21: a block (seasonal) maximum $\rightarrow$  corrected (page 8, line: 9)

14. p 7 l 22: The whole sentence with "Assumptions that our analyzed data..." needs rewriting.  $\rightarrow$  corrected (page 8, line 9-10)

15. p 7 l 26: there is something wrong with (1-1/T)th (you accidently inserted a comma)  $\rightarrow$  corrected (page 8, line 12)

16. p 7 1 33-35: please add a reference  $\rightarrow$  corrected (page 8, line: 23)

17. p 9 1 24: will presumably lead to a  $\rightarrow$  corrected (page 10, line 9)

18. p 9137: either from the model bias or from observational bias  $\rightarrow$  corrected (page 10, line 22)

19. p 10 l 4: high-resolution runs provide a more accurate representation  $\rightarrow$  corrected (page 10, line 26)

20. p 10 1 20: provided for the years  $\rightarrow$  we removed this sentence, since we no longer use the 1979-2005 simulation in the section 3.1

21. p 10 l 20: internal SST variations being in a different phase  $\rightarrow$  the same as above

22. p 11 1 18-21: reflect, generally after reductions, CO2 increase

23. Section 3.4: This one reads very well (also applies to section 2.3).

24. Figure 7: What is a fractional difference?  $\rightarrow$  We corrected it as a fractional change (difference in a relation to the mean in a reference period. (please see Fig 7 caption and text in page 13, line 10)

25. Figure 9: The caption talks about 50 years. As far as I understood the manuscript, shouldn't it be less years? Maybe I did not get, where the 50 years sample comes from?  $\rightarrow$  The sample comes from the 5-member ensemble of decadal runs. We've clarified it in the revised manuscript (Figure

9 caption).

26. Figure S2: There is either something wrong with the figure caption or there is a whole figure missing.  $\rightarrow$  We removed figure S2. The figure is not relevant for the analysis (section 3.1) in the current version, as it employs the 1979-2005 runs (instead 2006-2015). Because of the same reason we removed the old figure S3, which was showing the differences between the future and the 1996-2005 runs.

We would like to thank the reviewer for careful reading of this manuscript and for the constructive comments. We have responded to all comments below, added explanations where necessary, and applied most of the suggested changes or corrections.

We have also updated our analysis with new present climate runs, which were not available previously. Unlike the historical 1979-2005 simulation, the new runs now follow the HAPPI experiment protocol, i.e. they constitute an ensemble of decadal runs in 2006-2015. Therefore they are more suitable for the analysis investigating changes between the present and future climate. This approach is also more suitable to address the main reviewers concerns on the non-linear effects of the derived climate change, shown in the study. We will provide additional explanations below.

**Anonymous Referee #2**

A: Based on their analysis of the high resolution time slide experiments, the authors state that most of the changes due to anthropogenic forcing up to the 2C level only start occurring after the 1,5C threshold has been surpassed, and claim this may be due to non-linearities in the climate system (page 17, around line 30). While this may well be true to some point, I find it very hard to believe that while the climate change signal e.g. DJF MSLP and 850 hPa winds up to a 1,5C warmer world is practically zero (cf. Fig. S1a), we become a very strong response when we add the additional 0,5C (cf. Fig. S1b). The same could be argued for the other fields.

In my view, such a strong difference for such a small increment of external forcing could only come from (a) internal variability (for example, storminess is at a low level of its decadal variability in the period chosen for the 1,5C experiment, while it is on a high level on the period chosen for 2C) and/or (b) there are some issue in the set up of the high resolution simulations which have lead to these differences, and not the small change in the forcing. This does not turn the results per themselves wrong, but means that the authors may be misinterpreting (or at least over-interpreting) their results. In can be that the problem is related with the SST set up, as the authors shortly discuss in section 2.1. For example, how does the climate change signal look like for the (hopefully transient) lower resolution GCM simulations? How does the climate change signal look like for the single ensemble members? This would be important to analyse in detail to identify if the changes in precipitation and wind are continuous over time (plus natural decadal climate variability) or if indeed some strong "non-linear" effects occur. It is very improbable that the AMOC will collapse in a two degree warmer world. What might be important here is the thermodynamical effect (Clausius-Clapeyron), primarily for precipitation intensity. This is very important to rule out that the obtained results are not simply caused by some issue with the set up of the model simulations and clearly relate the changes to the increases of anthropogenic forcing.

AU: Thank you for the comment. Indeed, the "non-linear" effects of the derived changes, when comparing first half a degree warming (+1.5C minus present) with the additional half a degree warming (+2C minus +1.5C), are likely associated with the simulations set up. i.e. an asymmetry in the aerosol forcing between the present and future climate scenarios. The changes associated with warming at the 1.5°C level stem from an interplay of a number of forcings, including strong aerosol reductions, while an additional half a degree warming is solely a consequence of  $CO_2$  increases and ocean warming.

We enhanced the interpretation in the new revised version: Section 2.1: page 7, lines: 2-7 Section 3.1: page 10, lines: 35-37; page 11, lines: 1-9; page 12, lines: 10-21 Section 4: Page 17, lines 31-37 Minor comments:

**1 Page 2, line 34: regarding the role of cyclone clustering for the wind and flood impacts, the authors could refer for example to Priestley et al. (2017), which deals specifically with this topic.**

**AU: Thank you, we included the reference (Page 2, line 33)**

**2: Page 3, around line 10: The "poleward" shift of the storm tracks due to enhanced anthropogenic forcing is correct on zonal average, but this is not always true for Europe: while this is clear for the summer half year, the results for the winter, particularly DJF (the focus here), are different – see your own text in page 4, line 10. Please check e.g., the discussion in Zappa et al (2015), and also their Fig.1.**

AU: Yes, we agree that the "poleward shift" is a simplified statement. But it is a valid statement in the used context. We also added, that more discussion on the spatial changes of storms in different seasons will be included in latter part. (Page 3, line 12-13)

**3: Page 4, line 12: The results by Zappa et al. (2013b) do not contradict the results described in many of the papers mentioned in the previous sentence. For example, the results from Bengtsson et al. (2006), Pinto et al. (2009) or Zappa et al. (2013b) only show small differences in detail regarding the intensification of cyclone activity over the British Isles and the strong decrease in the Mediterranean. Thus, "however" is not the correct word here, maybe "Morevoer" would be more appropriate.**

**AU:** Thank you, we applied the correction. (Page 4, line 13)**

**4: Page 4/5: The authors describe their reasoning as if it would be the very first time that high-resolution global climate modelling is performed at \**

---

## Referee Report (RR1)

Many thanks for replying to my comments in great detail and for carefully revising the manuscript. The manuscript including the readability has improved substantially.

Almost all of my comments have been addressed. My comments regarding the scientific matter have been answered thoroughly to my satisfaction.

In my view, there are only some minor and technical revisions needed, before the manuscript is ready for publication.

**Minor point:**

To me, it is not entirely clear, how the ensembles were handled. From the section about the differences between the SLP, I would infer that given values in the text are referring to the average difference calculated in between all pairs of ensemble members. However, from the method section I would guess that all the calculations are based on ensemble averages. Either way of calculation has consequences for the calculation of spatially heterogenous variables such as wind speed and precipitation. Please state exactly how you used the 5-member ensembles to calculate differences and the statistics shown in the manuscript.

**Technical**

1) There are still references missing. For instance, I could not find Sillmann et al 2012, Seneviratne et al 2013, and Donat et al 2011 in the reference list. My guess is that there are more such incidences in the references, which I likely have overlooked. Please carefully check all the references in the text.

2) p5 l3: typo in Yang et al.

3) p6 l34 to p7 l1: sentences need some editing. The last word „for" does not make sense here. Assuming that the CO2 concentration is a weighted average, change the sentences to: „... concentrations, aerosols, ozone, land use, and land cover for the 1.5°C scenario. For the 2°C senario, these conditions are the same, except for the CO2 concentration, which is set to a weighted combination of the RCP2.6 and RCP4.5 scenarios."

4) p7 l10 change to: „The simulated features of large-scale circulation are compared with reanalysis data of monthly pressure..."

5) p9 l21 change to: „...which is 0.98. As shown below, the differences..."

6) p11 first word change to „experiments"

7) p11 lines 21, 23, 26: you are referring to Figure 4b, instead you mean Figure 4a as the sentences are about the circulation and not the precipitation

8) p12 l 34: „at 5% significance"; the figure caption says 10% significance. Which one is right?

9) Figure 1a: There are no contours showing differences between ERA Interim and NCEP/NCAR

10) Figure 1: There is no picture showing NCEP/NCAR as the caption says. Or is NCEP only shown as the difference-contours in Figure 1a? If so, the caption would need revising. Further, the numbers next to the contours almost vanish, even though they are sized appropriately. The issue here is that there are too few numbers occurring at the contour levels. The few that are present can easily be missed in the areas where contours and coastlines are dominant.

11) Figure 6: The caption says 10% significance level for Figures 6a and 6b, the text however says 5%. (see point 8)

12) Figure 8 caption: typo in the words DJF 95th daily wind percentiles. Remove the „)" between DJF and 95th.

13) Figure 9a) Please make the numbers next to the contour levels better readable.

14) Figure 9b) Make the caption consistent with the text. As you are analysing daily values, the number of occurences are number of days (as it is written in the text). Also, I noticed that the numbers at the contour levels differ substantially from the numbers given in the plot of the previous version of the manuscript. Why are the differences in between versions so large? Have you discovered an error in your analysis? Normally, I would not ask, but such a difference makes me wonder, which version is correct.

---

## Author Response (AR2)

We would like to thank the reviewer for careful reading of this manuscript and for the constructive comments. We have responded to all comments below, added explanations where necessary, and applied all the suggested corrections.

Many thanks for replying to my comments in great detail and for carefully revising the manuscript. The manuscript including the readability has improved substantially.
Almost all of my comments have been addressed. My comments regarding the scientific matter have been answered thoroughly to my satisfaction.
In my view, there are only some minor and technical revisions needed, before the manuscript is ready for publication.

**Minor point:**

To me, it is not entirely clear, how the ensembles were handled. From the section about the differences between the SLP, I would infer that given values in the text are referring to the average difference calculated in between all pairs of ensemble members. However, from the method section I would guess that all the calculations are based on ensemble averages. Either way of calculation has consequences for the calculation of spatially heterogenous variables such as wind speed and precipitation. Please state exactly how you used the 5-member ensembles to calculate differences and the statistics shown in the manuscript.

AU: Differences between the mean DJF climate were computed by comparing two, paired samples (present vs future runs, or future +1.5°C vs +2°C runs), each consisting of 50 seasonal (DJF) values. Statistical significance of derived differences in the mean DJF climate between future and present climate was additionally tested with the Wilcoxon signed rank test. We included more specific information in the Methods section (page 7, line 29-31). We also corrected some specifications for the storm tracking algorithm (page 8, line 36-37).

**Technical**

1) There are still references missing. For instance, I could not find Sillmann et al 2012, Seneviratne et al 2013, and Donat et al 2011 in the reference list. My guess is that there are more such incidences in the references, which I likely have overlooked. Please carefully check all the references in the text.
AU: Thank you, we checked all the references.

2) p5 l3: typo in Yang et al.  p5 l3
AU: Correction is applied.

3) p6 l34 to p7 l1: sentences need some editing. The last word „for" does not make sense here. Assuming that the CO2 concentration is a weighted average, change the sentences to: „... concentrations, aerosols, ozone, land use, and land cover for the 1.5°C scenario. For the 2°C senario, these

conditions are the same, except for the CO2 concentration, which is set to a weighted combination of the RCP2.6 and RCP4.5 scenarios."
AU: Thank you, we applied the correction.

4) p7 l10 change to: „The simulated features of large-scale circulation are compared with reanalysis data of monthly pressure..."
AU: Correction is applied.

5) p9 l21 change to: „...which is 0.98. As shown below, the differences..."
AU: Correction is applied.
6) p11 first word change to „experiments"
AU: Correction is applied. (p11, line 3)

7) p11 lines 21, 23, 26: you are referring to Figure 4b, instead you mean Figure 4a as the sentences are about the circulation and not the Precipitation.
AU: Yes, we corrected and inserted information on the figures we referred to.

8) p12 l 34: „at 5% significance"; the figure caption says 10% significance. Which one is right?
AU: Thank you, we corrected the value (10% significance)
9) Figure 1a: There are no contours showing differences between ERA Interim and NCEP/NCAR
AU: NCEP data is shown as difference –contours in Figure 1a. As mentioned in the text, the observational differences are in general much smaller than the differences between ERA and CAM5. Only in the vicinity of Greenland the observational differences are comparable with the 'ERA vs CAM5' differences. Therefore for the specified levels, the observational differences (NCEP –ERA) -contours appear only in this region.

10) Figure 1: There is no picture showing NCEP/NCAR as the caption says. Or is NCEP only shown as the difference-contours in Figure 1a? If so, the caption would need revising. Further, the numbers next to the contours almost vanish, even though they are sized appropriately. The issue here is that there are too few numbers occurring at the contour levels. The few that are present can easily be missed in the areas where contours and coastlines are dominant.

AU: Yes, NCEP data is shown only as difference –contours in Figure 1a. We corrected the caption. As suggested, we also increased density of the contour labels.

11) Figure 6: The caption says 10% significance level for Figures 6a and 6b, the text however says 5%. (see point 8)

AU: We applied correction

12) Figure 8 caption: typo in the words DJF 95th daily wind percentiles.

Remove the „)" between DJF and 95th.

AU: We applied correction

13) Figure 9a) Please make the numbers next to the contour levels better
readable.
AU: Thank you for the suggestion. We improved readability of contour levels in both Fig
9a and Fig 9b. We also added the information on the contour levels in the caption.

14) Figure 9b) Make the caption consistent with the text. As you are analysing
daily values, the number of occurences are number of days (as it is written
in the text). Also, I noticed that the numbers at the contour levels differ
substantially from the numbers given in the plot of the previous version of
the manuscript. Why are the differences in between versions so large?
Have you discovered an error in your analysis? Normally, I would not ask,
but such a difference makes me wonder, which version is correct

AU: The differences in the presented scale of values stem only from using different units.
As we mentioned in the revision, Figure 9b in the new version shows numbers per decade
[number /decade]. In contrast, statistics in the earlier versions were presented as numbers
per 50 years [number /five decades].

Also, both Figure 9a and b focuses on the 3-hr values of the extracted storm tracks, and
not daily values. We corrected the text (page 15, line: 25) as "3-h storm occurrences".
Thank you for all the suggestions and corrections.

[revised manuscript text omitted]